# POSE-RFT: ALIGNING MLLMS FOR 3D POSE GENERATION VIA HYBRID ACTION REINFORCEMENT FINE-TUNING

**Bao Li**[1,2*]**, Xiaomei Zhang**[1,2*]**, Miao Xu**[1,3]**, Zhaoxin Fan**[4]**, Xiangyu Zhu**[1,2]**, Zhen Lei**[1,2,3†]

[1]MAIS, Institute of Automation, Chinese Academy of Sciences
[2]School of Artificial Intelligence, University of Chinese Academy of Sciences
[3]CAIR, HKISI, Chinese Academy of Sciences
[4]Beijing Advanced Innovation Center for Future Blockchain and Privacy Computing,
School of Artificial Intelligence, Beihang University
{libao2023, zhangxiaomei2016, xiangyu.zhu, zhen.lei}@ia.ac.cn
{miao.xu}@cair-cas.org.hk, {zhaoxinf}@buaa.edu.cn

## ABSTRACT

Generating 3D human poses from multimodal inputs such as text or images requires models to capture both rich semantic and spatial correspondences. While pose-specific multimodal large language models (MLLMs) have shown promise, their supervised fine-tuning (SFT) paradigm struggles to resolve the task's inherent ambiguity. Its reliance on objectives like SMPL parameter regression creates a critical alignment gap, compromising the model's ability to achieve the required semantic and spatial fidelity. To close the gap, we propose Pose-RFT, a framework that shifts the learning paradigm from supervised imitation to reward-driven reinforcement fine-tuning (RFT). We address the core technical challenge of this task: a hybrid action space requiring joint optimization of discrete language and continuous pose outputs. To this end, we introduce HyGRPO, a hybrid reinforcement learning algorithm that enables stable optimization by performing group-wise reward normalization over sampled responses. Pose-RFT incorporates task-specific reward functions to guide optimization towards spatial alignment in image-to-pose generation and semantic consistency in text-to-pose generation. Extensive experiments on multiple pose generation benchmarks demonstrate that Pose-RFT significantly improves performance over existing pose-specific MLLMs, validating the effectiveness of our approach in closing the alignment gap for 3D pose generation.

## 1 INTRODUCTION

Recent advances in 3D human pose generation (Delmas et al., 2022; 2023; 2024; Miao et al., 2024; Wang et al., 2025b; Tevet et al., 2022; Li et al., 2025a) have increasingly focused on addressing the problem of understanding and reasoning about 3D human poses from multimodal inputs, such as images and text. Among these, pose-specific multimodal large language models (MLLMs) (Feng et al., 2024; Lin et al., 2024; Li et al., 2025c) have emerged as a promising direction, extending general-purpose language models with dedicated pose decoders to enable joint reasoning over language, vision, and 3D pose. While these models have shown strong performance, their standard training via supervised fine-tuning (SFT) exposes a fundamental limitation.

The central challenge is the inherent one-to-many nature of 3D pose generation. This ambiguity is explicit in the text-to-pose task (Delmas et al., 2022; Li et al., 2024b), where a single prompt can map to a broad distribution of valid poses. It is implicit in the image-to-pose task (Von Marcard et al., 2018; Shen et al., 2023; Yan et al., 2024; Wang et al., 2025a), a classic ill-posed problem where multiple plausible 3D poses can yield the same 2D evidence. The SFT paradigm, which learns a deterministic mapping via regression to a single ground truth for each sample, is fundamentally

---

*Equal contribution.
†Corresponding author.

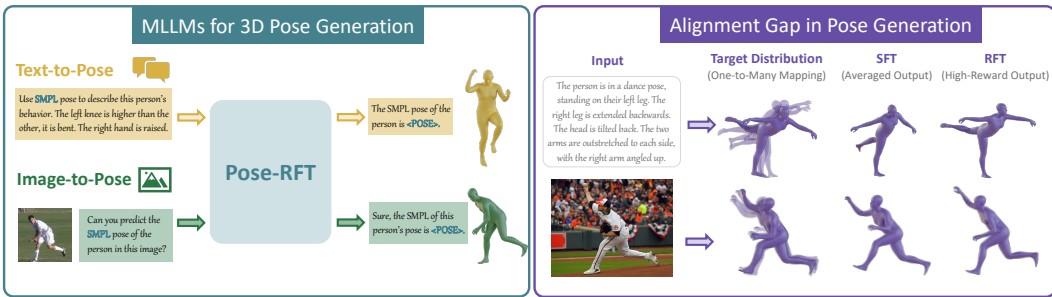

Figure 1: **Examples and Motivation. Left:** An overview of our Pose-RFT framework for multimodal 3D pose generation. **Right:** Illustrating the alignment gap. While SFT yields a suboptimal averaged output, RFT produces a high-reward output for superior semantic and spatial alignment.

misaligned with this property. Consequently, SFT models are driven to predict an averaged, often suboptimal, output to minimize expected error across the dataset. As illustrated in Figure 1, this creates a critical alignment gap between the model's predictions and the true objectives of semantic consistency and spatial accuracy.

To close this alignment gap, a paradigm shift from supervised imitation to goal-driven optimization is necessary. Reinforcement Learning (RL) (Schulman et al., 2017; Rafailov et al., 2023) offers a principled framework for this, enabling models to learn from reward signals that directly reflect task-specific goals. However, applying RL to this domain presents a significant challenge: most existing Reinforcement Fine-Tuning (RFT) algorithms (Achiam et al., 2023; Jaech et al., 2024; Liu et al., 2025b; Shen et al., 2025; Li et al., 2025b) are designed for the discrete token spaces of language and are not equipped to handle the fine-grained, continuous parameters of 3D human poses.

To address these challenges, we propose Pose-RFT, a novel reinforcement fine-tuning framework specifically designed for 3D human pose generation in MLLMs (see Figure 2). **First**, we formulate the task as a hybrid action space reinforcement learning problem, where the policy must simultaneously produce discrete actions (e.g., text tokens) and continuous actions (e.g., 3D pose parameters). To effectively manage the inherent uncertainty and enable the stochastic exploration required by RL, the continuous action is modeled by a multivariate Gaussian policy. This policy is parameterized by a dedicated pose head that predicts both the mean and covariance for a given state. **Second**, we introduce HyGRPO, a novel online hybrid reinforcement learning algorithm designed to achieve stable optimization in the challenging hybrid action space. By leveraging group-wise reward normalization over multiple sampled outputs, HyGRPO directly optimizes the policy, effectively steering it towards high-reward responses. **Third**, we propose four task-specific reward functions to guide policy optimization: (i) a spatial location reward for image-to-pose generation, (ii) a semantic alignment reward for text-to-pose generation, (iii) a format correctness reward, and (iv) a text embedding similarity reward. By training with diverse outputs and structured feedback, HyGRPO encourages the model to generate 3D poses that are spatially accurate and semantically aligned.

In summary, our main contributions are as follows:

(1) We propose Pose-RFT, the first reinforcement fine-tuning framework specifically designed for 3D human pose generation in MLLMs. (2) We develop HyGRPO, a hybrid-action reinforcement learning algorithm that effectively optimizes both discrete and continuous outputs in pose-specific MLLMs. (3) Extensive experiments on multiple pose generation benchmarks demonstrate that Pose-RFT significantly improves performance over existing pose-specific MLLMs, validating the effectiveness of hybrid action reinforcement fine-tuning for 3D pose generation.

## 2 RELATED WORK

### 2.1 HUMAN POSE GENERATION

Human pose generation involves producing 3D human poses conditioned on either images or text. For image-to-pose generation, also known as pose estimation, existing approaches are typically divided into optimization-based and regression-based methods. Optimization-based methods (Bogo et al., 2016; Pavlakos et al., 2019) estimate 3D pose parameters by aligning projected

joints with detected 2D keypoints through iterative refinement. In contrast, regression-based approaches (Kanazawa et al., 2018; Cai et al., 2023; Dwivedi et al., 2024; Goel et al., 2023) rely on deep neural networks to directly predict 3D poses from input images. Text-to-pose generation aims to synthesize 3D human poses based on textual descriptions, such as physical attributes or actions (Delmas et al., 2022; Tevet et al., 2022; Hong et al., 2022). Although these methods have shown promising results, they remain confined to either image-to-pose or text-to-pose generation, without a unified framework capable of leveraging cross-modal knowledge to infer human poses from both visual and textual inputs.

## 2.2 MULTIMODAL LARGE LANGUAGE MODELS

Multimodal Large Language Models (MLLMs) (Achiam et al., 2023; Liu et al., 2023; Li et al., 2024a; Wang et al., 2024b; Chen et al., 2024b) have shown strong performance in vision-language understanding tasks by jointly modeling visual inputs and natural language. These models excel at multimodal reasoning, visual grounding, and instruction following, enabling them to comprehend complex visual content in diverse application scenarios. Leveraging these capabilities, recent works have successfully applied MLLMs to downstream vision-centric tasks such as image segmentation (Lai et al., 2024; Bai et al., 2024), anomaly detection (Gu et al., 2024), and keypoint localization (Wang et al., 2024a), demonstrating their transferability beyond purely linguistic domains.

To adapt MLLMs to downstream tasks, post-training strategies such as supervised fine-tuning (SFT) and reinforcement fine-tuning (RFT) are commonly used. Recent efforts such as ChatPose (Feng et al., 2024) and UniPose (Li et al., 2025c) have applied SFT to extend MLLMs for 3D pose generation, leveraging their vision-language reasoning capabilities. However, these methods rely solely on SFT and do not incorporate reinforcement-based optimization. The absence of RFT limits the model's capacity to further refine generation quality, particularly in scenarios involving ambiguity and task-specific alignment.

## 2.3 REINFORCEMENT LEARNING

Reinforcement learning (RL) (Sutton et al., 1998) is a core paradigm in machine learning, where an agent learns a policy-a mapping from observations to actions-by interacting with an environment and optimizing cumulative rewards. Through trial-and-error learning, the agent improves its policy based on feedback in the form of scalar rewards. Classical algorithms such as Q-learning(Watkins & Dayan, 1992) have been successfully applied in fields such as robotics, autonomous control, and game playing. With the rise of large language models (Radford et al., 2018; Touvron et al., 2023; Achiam et al., 2023), Reinforcement Learning with Human Feedback (RLHF) (Bai et al., 2022) has become as a key technique for fine-tuning models using human preference data. RLHF leverages algorithms like Proximal Policy Optimization (PPO) (Schulman et al., 2017) and Direct Preference Optimization (DPO) (Rafailov et al., 2023) to guide model behavior for improving the alignment, coherence, and helpfulness in response generation.

In the context of multimodal large language models, recent works (Zhou et al., 2025; Liu et al., 2025a; Zhan et al., 2025; Liu et al., 2025b; Yang et al., 2025; Zhang et al., 2025; Shen et al., 2025) have explored the use of RL with verifiable reward signals to enhance visual reasoning. However, the application of RL to 3D human pose generation remains underexplored, primarily due to the continuous nature of pose regression, which poses challenges for RL methods originally designed for discrete action spaces. To address similar challenges in other domains, several works have proposed hybrid discrete-continuous action formulations (Lowe et al., 2017; Fan et al., 2019; Li et al., 2021), offering a promising direction for adapting reinforcement learning to structured continuous tasks such as 3D pose generation.

## 3 METHODOLOGY

This section first reformulates 3D human pose generation as a reinforcement learning problem under a hybrid action space. It then introduces the proposed *Hybrid Action Space Group Relative Policy Optimization* (HyGRPO) algorithm, which jointly optimizes discrete language and continuous pose outputs. Finally, it describes how HyGRPO is applied to fine-tune pose-specific MLLMs using task-specific reward functions designed for 3D human pose generation.

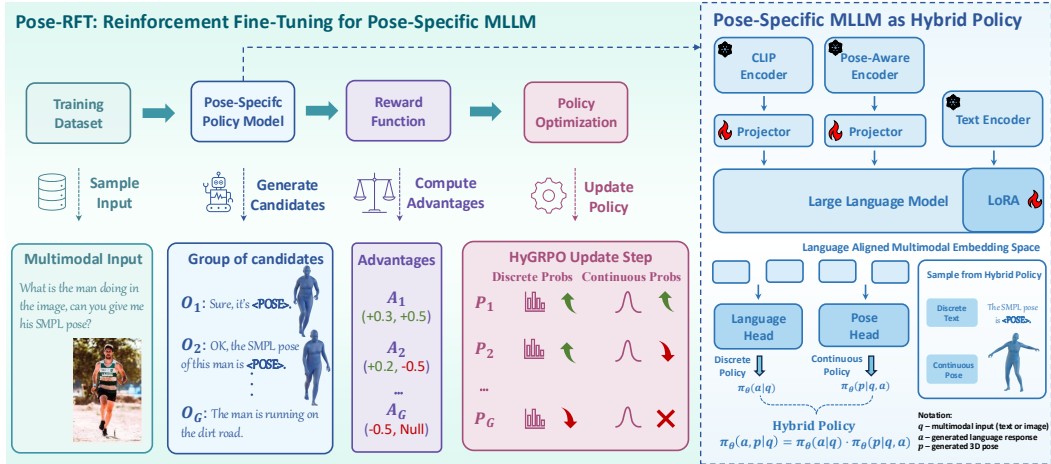

Figure 2: **Overview of Pose-RFT Framework.** Our reinforcement fine-tuning framework for pose-specific MLLMs. Given a multimodal input, the model generates multiple hybrid responses (text + pose). These candidates are evaluated using task-specific rewards, and our HyGRPO algorithm updates the policy to promote the generation of higher-reward outputs.

## 3.1 REFORMULATING 3D POSE GENERATION FOR REINFORCEMENT LEARNING

We formulate the 3D human pose generation within pose-specific MLLMs as a hybrid action reinforcement learning problem. The model operates in a hybrid action space comprising discrete language tokens and continuous 3D poses. The overall policy is defined as:

$$\pi_\theta(a, p|q) = \pi_\theta(a|q) \cdot \pi_\theta(p|q, a), \tag{1}$$

where $q$ denotes a multimodal input, $a$ a represents discrete textual responses, and $p$ denotes continuous 3D pose parameters. We regard the joint distribution $\pi_\theta(a, p|q)$ as the overall policy, which is factorized into a discrete sub-policy $\pi_\theta(a|q)$ modeling the distribution over textual responses, and a continuous sub-policy $\pi_\theta(p|q, a)$ modeling the distribution over 3D poses conditioned on both the input query and the generated language response.

To parameterize the continuous policy, we adopt a multivariate Gaussian distribution defined over the space of 3D human poses:

$$\pi_\theta(p|q, a) = \mathcal{N}(p; \mu_\theta(q, a), \Sigma_\theta(q, a)), \tag{2}$$

where the mean $\mu_\theta(q, a)$ and covariance $\Sigma_\theta(q, a)$ are predicted by a continuous pose head conditioned on the multimodal input $q$ and the discrete response $a$. This probabilistic formulation captures the aleatoric uncertainty inherent in 3D human pose generation by modeling the conditional distribution over continuous pose vectors. Furthermore, the use of a differentiable multivariate Gaussian enables both stochastic sampling during training and efficient gradient-based optimization within the continuous pose space.

Benefiting from the strong cross-modal alignment established during MLLM pretraining, both the discrete and continuous policies are built on a shared language-aligned multimodal embedding space. To further enhance this representation, we augment the MLLM with a pose-aware encoder, a Vision Transformer pretrained on pose estimation, to extract more informative and pose-relevant visual features. These features are fused with the language-aligned multimodal embeddings to yield a more informative and pose-relevant state space for reinforcement learning. Details of the pose-aware encoder and the visual fusion strategy are provided in Appendix A. Taken together, the architectural enhancement from the pose-aware encoder and the probabilistic modeling from the Gaussian policy establish a powerful and robust baseline model, which we then elevate further using our reinforcement fine-tuning framework.

## 3.2 HYGRPO: HYBRID ACTION SPACE GROUP RELATIVE POLICY OPTIMIZATION

To optimize the hybrid policy defined in the previous section, we introduce Hybrid Action Space Group Relative Policy Optimization (HyGRPO), an online reinforcement learning algorithm designed to enhance pose-specific MLLMs for 3D human pose generation. HyGRPO is designed to

operate directly on the hybrid action space, jointly optimizing the discrete language and continuous pose heads within the shared MLLM embedding space. By doing so, it facilitates coherent alignment between the model's textual and pose outputs, effectively bridging the gap between standard discrete token prediction and continuous parameter generation.

To handle hybrid outputs, HyGRPO models the policy $\pi_\theta$ over both discrete text answers $a$ and continuous human poses $p$ conditioned on input question $q$. For each training sample $q$ from dataset $\mathcal{D}$, we sample $G$ output candidates $\{a_i, p_i\}_{i=1}^{G} \sim \pi_\theta(\cdot|q)$, and optimize the policy using the following objective:

$$\mathbb{E}_{q\sim\mathcal{D},\{a_i,p_i\}_{i=1}^{G}\sim\pi_\theta(\cdot|q)} \left[ \frac{1}{G} \sum_{i=1}^{G} r_i(\theta)\hat{A}_i \right], \tag{3}$$

where $r_i(\theta)$ is the importance weight of the $i$-th sampled output, computed as the ratio between the current policy and the reference policy:

$$r_i(\theta) = \frac{\pi_\theta(a_i, p_i|q)}{\pi_{\text{ref}}(a_i, p_i|q)} = \underbrace{\frac{\pi_\theta(a_i|q)}{\pi_{\text{ref}}(a_i|q)}}_{r_d(a_i|q)} \cdot \underbrace{\frac{\pi_\theta(p_i|q, a_i)}{\pi_{\text{ref}}(p_i|q, a_i)}}_{r_c(p_i|q,a_i)}. \tag{4}$$

A key design choice in HyGRPO is the decomposition of the normalized advantage, $\hat{A}$, into discrete and continuous components, which measure the quality of the textual response and the predicted pose, respectively:

$$\hat{A}(q, a, p) = \underbrace{\hat{F}(q, a)}_{\text{discrete advantages}} + \underbrace{\hat{\Delta}(q, a, p)}_{\text{continous advantages}}, \tag{5}$$

where $\hat{F}$ measures the quality of the generated textual response, and $\hat{\Delta}$ evaluates the predicted pose quality. To ensure training stability, we adopt clipped importance sampling as in PPO (Schulman et al., 2017). The final HyGRPO training objective is:

$$\begin{aligned} \mathcal{J}_{\text{HyGRPO}} = & \mathbb{E}_{q\sim\mathcal{D},\{a_i,p_i\}_{i=1}^{G}\sim\pi_\theta(\cdot|q)} \left[ \frac{1}{G} \sum_{i=1}^{G} \Big( \min(r_d\hat{F}_i, \text{clip}(r_d, 1-\epsilon, 1+\epsilon)\hat{F}_i) \Big) \right. \\ & \left. + \frac{1}{V} \sum_{i=1}^{V} \Big( \min(r_c\hat{\Delta}_i, \text{clip}(r_c, 1-\epsilon, 1+\epsilon)\hat{\Delta}_i) \Big) - \beta D_{\text{KL}}(\pi_\theta\|\pi_{\text{ref}}) \right], \end{aligned} \tag{6}$$

where $G$ is the number of generated candidates, $V$ is the subset of candidates that include a valid pose output. This objective provides separate, targeted advantage signals for the discrete and continuous policy heads, enabling stable and generalizable training across the hybrid action space. The full derivation and algorithm are detailed in Appendix B and C.

### 3.3 GUIDING HyGRPO WITH TASK-SPECIFIC REWARDS

The HyGRPO algorithm is guided by a suite of task-specific reward functions designed on a core principle: each reward provides targeted feedback for a distinct component of the MLLM's hybrid output. This modular design ensures comprehensive supervision, governing not only the continuous pose outputs (for spatial and semantic accuracy) but also the discrete textual outputs (for conversational format and correctness). This approach is crucial as it enhances the model's new pose generation capabilities while preserving its foundational conversational capabilities.

**Spatial Location Reward in Image-to-Pose Generation.** In the image-to-pose generation task, the model is expected to output SMPL pose coefficients conditioned on the input image. To encourage spatial accuracy, the reward should reflect how well the predicted pose aligns with the visual input. Following standard 3D human pose estimation practice, this reward is defined as the inverse of the mean joint position error—the Euclidean distance between predicted and ground-truth 3D joint locations:

$$\mathcal{R}_{\text{joint}} = \frac{1}{||J_{\text{pred}} - J_{\text{gt}}||_2}. \tag{7}$$

**Semantic Alignment Reward in Text-to-Pose Generation.** In the text-to-pose generation task, the model is expected to predict SMPL pose coefficients conditioned on a text prompt. Unlike image-to-pose generation, which emphasizes joint-level accuracy, this task focuses on high-level semantic alignment between the textual description and the generated pose.

To quantify this alignment, we adopt a pretrained text-pose retrieval model that maps both inputs into a shared embedding space. Specifically, the retrieval model comprises a text encoder $\phi_t(\cdot)$ and a pose encoder $\phi_p(\cdot)$, both projecting their respective inputs into a shared embedding space. The semantic alignment reward is defined as the similarity score between the encoded text and the generated pose:

$$\mathcal{R}_{\text{semantic}} = \cos(\phi_t(q), \phi_p(p)). \tag{8}$$

**Format Reward.** To encourage the model to generate responses that conform to a specified format, we introduce a format reward, denoted as $R_{\text{format}}$. For instance, we expect the model to produce outputs enclosed in a template such as: *"The SMPL pose of this person is <POSE>."* To enforce this constraint, we apply regular expression matching to assess format compliance. The format reward is defined as:

$$\mathcal{R}_{\text{format}} = \begin{cases} 1, & \text{if the output matches the expected format} \\ 0, & \text{otherwise} \end{cases}. \tag{9}$$

**Text Embedding Similarity Reward.** To preserve general QA capabilities while fine-tuning for pose-centric tasks, we incorporate a text reward that encourages semantic agreement between generated and ground-truth answers in vision-language QA tasks. Specifically, we utilize the BGE-M3 encoder (Chen et al., 2024a) to compute dense embeddings for both the model-generated answer and the ground-truth response. The reward is defined as the cosine similarity between the normalized embeddings of the predicted and ground-truth answers:

$$\mathcal{R}_{\text{text}} = \cos(E(a_{\text{pred}}), E(a_{\text{gt}})). \tag{10}$$

## 4 EXPERIMENTS

### 4.1 EXPERIMENTAL SETUP

**Datasets.** To train Pose-RFT, we incorporate four types of data sources to enhance multimodal understanding: **(1) Text-Pose Data.** We utilize the PoseScript dataset (Delmas et al., 2022), which provides natural language descriptions paired with 3D human poses. This enables the model to learn fine-grained semantic correlations between language and human poses. **(2) Image-Pose Data.** Following prior works (Goel et al., 2023; Feng et al., 2024; Li et al., 2025c), we adopt standard human pose estimation training datasets, including Human3.6M (Ionescu et al., 2013), MPI-INF-3DHP (Mehta et al., 2017), COCO (Lin et al., 2014), and MPII (Andriluka et al., 2014). For evaluation, we use the 3DPW (Von Marcard et al., 2018) and Human3.6M test sets. **(3) Image-Text Data.** We employ the BEDLAM-Script dataset introduced in PoseEmbroider (Delmas et al., 2024), a curated multimodal dataset containing images, 3D poses, and textual descriptions, constructed based on the BEDLAM dataset (Black et al., 2023). **(4) VQA Data.** For visual question answering, we utilize the LLaVA-Instruct-150k dataset (Liu et al., 2023).

**Metrics.** We evaluate our model on both image-to-pose and text-to-pose tasks using reconstruction and retrieval metrics. **Image-to-Pose Reconstruction Metrics:** We report the Mean Per Joint Position Error (MPJPE) and the Procrustes-aligned MPJPE (PA-MPJPE), which measure the average Euclidean distance between predicted and ground-truth joint positions, with and without Procrustes alignment, respectively. **Text-to-Pose Retrieval Metrics:** Following (Feng et al., 2024; Lin et al., 2024; Li et al., 2025c), we report Recall@K (K = 5, 10, 20) for both text-to-pose ($R^{T2P}$) and pose-to-text ($R^{P2T}$) retrieval tasks, which assess the accuracy of matching poses with their corresponding textual descriptions.

**Implementation Details.** We adopt LLaVA-1.5V-7B (Liu et al., 2023) as the vision-language backbone. For the pose-aware encoder, we employ the pretrained Vision Transformer from HMR2.0

Table 1: **Comparison on Human Pose Estimation task.** Reconstruction errors are reported on the 3DPW and Human3.6M datasets.

| Method | 3DPW | | Human3.6M | | RPE | |
|---|---|---|---|---|---|---|
| | MPJPE | PA-MPJPE | MPJPE | PA-MPJPE | MPJPE | PA-MPJPE |
| HMR (Von Marcard et al., 2018) | 130.0 | 76.7 | 88.0 | 56.8 | - | - |
| SPIN (Kolotouros et al., 2019) | 96.9 | 59.2 | 62.5 | 41.1 | 244.9 | 107.3 |
| PyMAF (Zhang et al., 2021) | 92.8 | 58.9 | 57.7 | 40.5 | - | - |
| HMR2.0 (Goel et al., 2023) | 70.0 | 44.5 | **44.8** | 33.6 | 225.2 | 105.7 |
| MEGA (Fiche et al., 2024) | **67.5** | **41.0** | - | - | - | - |
| TokenHMR (Dwivedi et al., 2024) | 71.0 | 44.3 | - | - | - | - |
| ChatPose (Feng et al., 2024) | 163.6 | 81.9 | 126.0 | 82.4 | 275.0 | 101.8 |
| UniPose (Li et al., 2025c) | 94.7 | 59.1 | 69.2 | **41.8** | 213.4 | 94.1 |
| Pose-RFT (Ours) | **85.9** | **51.6** | **63.0** | 44.5 | **198.6** | **87.0** |

Table 2: **Comparison on Text-to-Pose Generation Task.** Retrieval metrics (Recall@K, K=5, 10, 20) are reported on the PoseScript dataset under two evaluation protocols.

| Method | PoseScript (Full Retrieval) | | PoseScript (Random Sampling) | |
|---|---|---|---|---|
| | $R^{T2P}$ ↑ | $R^{P2T}$ ↑ | $R^{T2P}$ ↑ | $R^{P2T}$ ↑ |
| PoseScript (Delmas et al., 2022) | 40.4 52.3 65.0 | 41.4 54.1 65.9 | 73.3 82.5 89.4 | 70.0 82.5 87.4 |
| ChatPose (Feng et al., 2024) | 17.6 25.3 35.8 | 28.0 39.0 54.4 | 39.9 50.6 58.7 | 56.1 65.3 72.5 |
| ChatHuman (Lin et al., 2024) | 41.8 52.6 65.1 | 42.1 52.3 66.5 | - - - | - - - |
| UniPose (Li et al., 2025c) | - - - | - - - | **73.7** 82.4 **89.6** | 70.9 80.5 89.6 |
| Pose-RFT (Ours) | **42.2 53.0 65.5** | **45.3 57.2 70.4** | 71.8 **82.6** 88.7 | **74.6 86.5 91.5** |

(Goel et al., 2023). Reinforcement fine-tuning follows the settings of Visual-RFT (Liu et al., 2025b) and VLM-R1 (Shen et al., 2025). During both pretraining and fine-tuning, the CLIP encoder and the pose-aware encoder are kept frozen, while the projector and task head are updated. The large language model is fine-tuned using LoRA (Hu et al., 2022). Further implementation details are provided in the Appendix D.

## 4.2 COMPARISONS ON HUMAN POSE GENERATION TASKS

**Image-to-Pose Generation.** We evaluate our method on both standard reconstruction and complex reasoning tasks for image-to-pose generation. On standard benchmarks like 3DPW (Von Marcard et al., 2018) and Human3.6M (Ionescu et al., 2013), Table 1 shows that Pose-RFT significantly outperforms other MLLM-based approaches, demonstrating the efficacy of our reinforcement fine-tuning framework in closing the alignment gap. While a performance gap remains compared to traditional specialist models in this setting, the unique advantage of the MLLM paradigm is evident on the Reasoning Pose Estimation (RPE) task (Feng et al., 2024). On this more complex benchmark, which requires visual-language reasoning that specialist models cannot perform, Pose-RFT establishes a new state-of-the-art, validating the effectiveness of our MLLM-based approach for advanced, reasoning-driven pose estimation.

**Text-to-Pose Generation.** We compare Pose-RFT with existing text-conditional pose generation models (Delmas et al., 2022; Feng et al., 2024; Lin et al., 2024; Li et al., 2025c) on PoseScript-H2 test set (Delmas et al., 2022). Following the standard protocol for this task, we generate 3D poses from text prompts and then use a pretrained retrieval model (Delmas et al., 2022) to compute Recall@K scores as a proxy for generation quality. To ensure a comprehensive comparison, we report results under two established evaluation protocols (Full Retrieval and Random Sampling). As shown in Table 2, Pose-RFT achieves the best performance across most metrics. We attribute this success to our reinforcement fine-tuning with a semantic alignment reward, which effectively enhances the model's ability to capture fine-grained text-pose correspondence.

## 4.3 Ablation Studies and Discussions

**Pose-Aware Encoder.**    We evaluate the effectiveness of the proposed pose-aware encoder, our specialized visual module designed to capture fine-grained pose information from the input image. As shown in Figure 3, compared to relying solely on a generic CLIP encoder, this specialized module leads to a significantly higher spatial location reward score on the 3DPW dataset. This improvement establishes a more powerful SFT baseline for subsequent fine-tuning. We do observe, however, that this vision-centric module brings little benefit to the semantic reward in the text-to-pose task, an expected outcome as its features are less aligned with textual inputs.

**Distributional Modeling.**    Next, we analyze the impact of modeling the 3D pose output as a probabilistic distribution rather than a deterministic one–a key prerequisite for our sampling-based RFT approach. As shown in Table 3, introducing the distributional head (Baseline + Dist.) by itself yields only marginal changes in performance compared to the deterministic (Baseline). However, its crucial role is revealed when combined with reinforcement learning (Baseline + Dist. + RFT), where it achieves the best performance. This synergy demonstrates that distributional modeling is a critical enabler, facilitating more effective reward-driven exploration and learning.

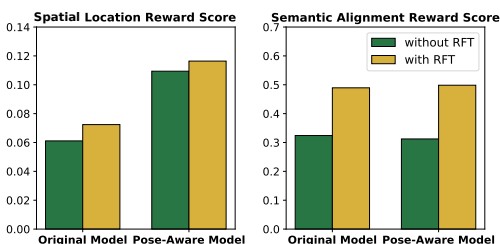

Figure 3:  Ablation Study of Pose-RFT's Core Components. Both the Pose-Aware Encoder and Reinforcement Fine-tuning (RFT) contribute positively, with RFT providing the most significant gains across both semantic and spatial rewards.

**Effectiveness of Reinforcement Fine-tuning.**    With a stronger architecture and a probabilistic policy in place, we now demonstrate the core contribution of our work. As shown in both Figure 3 and Table 3, applying reinforcement fine-tuning (+ RFT) provides the most significant performance gains across all tasks and metrics. This consistently positive result validates our central hypothesis: shifting from a supervised paradigm to a reward-driven RFT paradigm is highly effective in enhancing the alignment between language, vision, and 3D pose representations, successfully closing the alignment gap.

Table 3: Ablation study on distributional modeling (denoted as "Dist.") for 3D pose generation. Reconstruction and retrieval metrics are reported on the 3DPW and PoseScript-H2 datasets.

| Method | Dist. | RFT | Image-to-Pose Generation | | Text-to-Pose Generation | |
|---|---|---|---|---|---|---|
| | | | MPJPE $\downarrow$ | PA-MPJPE $\downarrow$ | mRecall$^{T2P}$ $\uparrow$ | mRecall$^{P2T}$ $\uparrow$ |
| Baseline | ✗ | ✗ | 90.4 | 57.1 | 36.2 | 41.5 |
| Baseline + Dist. | ✓ | ✗ | 91.4 | 59.2 | 37.4 | 42.0 |
| Baseline + Dist. + RFT | ✓ | ✓ | 85.9 | 51.6 | 53.6 | 57.6 |

**Necessity of Hybrid-Action RL (HyGRPO).**    Finally, we validate that a specialized hybrid-action algorithm is essential for this success. We compare our proposed HyGRPO with GRPO, an algorithm designed for discrete-only action spaces. As illustrated in Figure 4, GRPO fails to improve the quality of the continuous 3D pose generation. In contrast, HyGRPO, by jointly optimizing both discrete token and continuous pose parameters under the guidance of task-specific rewards, yields consistent and substantial improvements. This confirms that our novel HyGRPO algorithm is the key technical component that enables the success of RFT in this complex, hybrid-action task.

## 4.4 Qualitative Results

We provide qualitative results to visually validate our method on both tasks. For text-to-pose, Figure 5 illustrates the training progression, showing a clear improvement in semantic alignment and structural plausibility as our reinforcement fine-tuning proceeds. For image-to-pose, Figure 6 demonstrates the superior spatial accuracy and realism of our Pose-RFT in a direct comparison against other leading MLLM-based methods on challenging in-the-wild images.

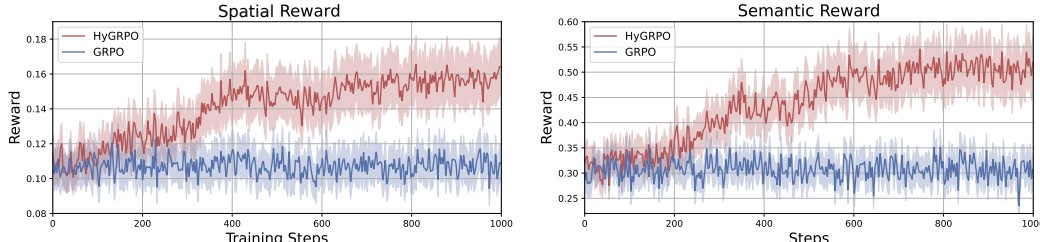

Figure 4: **Comparison between GRPO and HyGRPO.** Training reward curves for pose generation. The discrete-only GRPO fails to yield improvements, whereas our proposed HyGRPO achieves consistent gains, demonstrating that a hybrid-action approach is essential for optimizing continuous pose outputs.

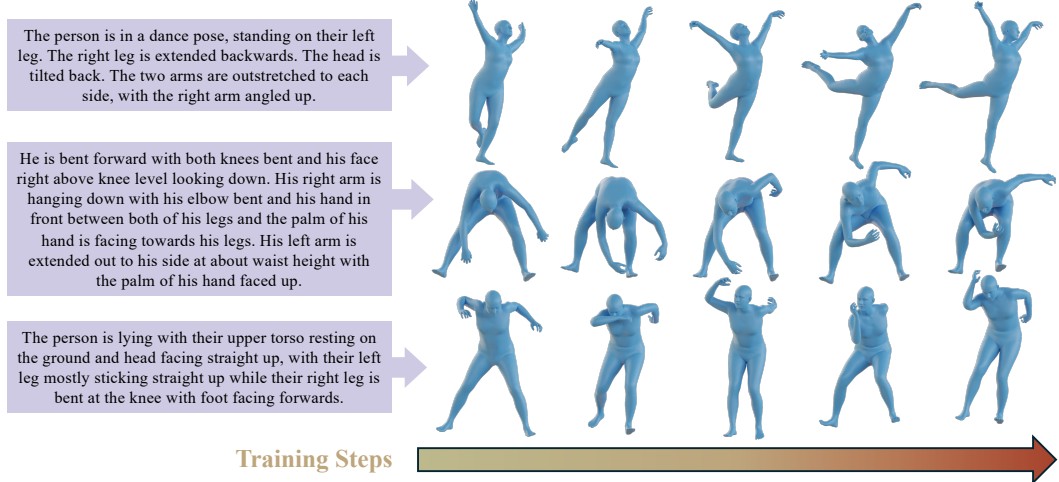

Figure 5: **Training progression of text-to-pose generation.** As reinforcement fine-tuning progresses (left to right), 3D poses generated from fixed text prompts exhibit increasingly improved semantic consistency and structural plausibility.

## 5 CONCLUSION

In this paper, we presented Pose-RFT, the first reinforcement fine-tuning framework designed to resolve the critical alignment gap in MLLM-based 3D pose generation. We attribute this gap to the fundamental mismatch between the standard SFT paradigm and the inherent one-to-many nature of the pose generation task. To address this, Pose-RFT introduces a paradigm shift to RFT, enabled by our novel HyGRPO algorithm, which is specifically designed to handle the challenging discrete-continuous hybrid action space. Guided by task-specific rewards for spatial accuracy and semantic alignment, our framework directly optimizes for the true objectives of the task. Extensive experiments on multiple benchmarks demonstrate that Pose-RFT consistently outperforms existing pose-specific MLLMs. These findings validate that our hybrid-action reinforcement fine-tuning approach is an effective method for closing the alignment gap. More broadly, this work highlights the significant potential of RFT for unlock-

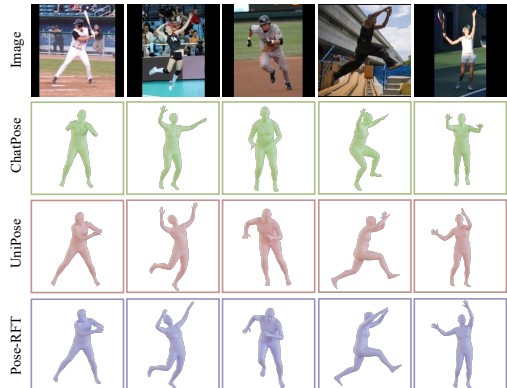

Figure 6: Qualitative comparison on image-to-pose generation. Our Pose-RFT (bottom row) exhibits superior spatial accuracy and realism over baselines, especially in capturing challenging limb orientations and overall dynamics.

ing the full capabilities of MLLMs on complex, ambiguous generation tasks, paving the way for more aligned and controllable human-centric content generation.

ACKNOWLEDGMENTS

This work was supported in part by the National Key Research & Development Program (No. 2025ZD0123501), the Chinese National Natural Science Foundation Projects (Nos. 62206280, 92570119), the Beijing Natural Science Foundation (No. 4254100), the Beijing Advanced Innovation Center for Future Blockchain and Privacy Computing, the Young Scientists Fund of the State Key Laboratory of Multimodal Artificial Intelligence Systems (ES2P100113), the Postdoctoral Fellowship Program and China Postdoctoral Science Foundation (Nos. 2024M764093, BX20250485), and the InnoHK program.

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

## A DETAILS OF POSE-AWARE ENCODER AND THE VISUAL FUSION STRATEGY

This section provides additional implementation details of the Pose-Aware Encoder and the corresponding visual feature fusion strategy used in our framework.

Previous works (Liu et al., 2023; Feng et al., 2024) typically adopt the CLIP visual encoder (Radford et al., 2021) as the visual backbone. However, since CLIP is pretrained using global and coarse-grained supervision from image-caption pairs, it struggles to capture fine-grained pose details. In contrast, pose estimation tasks require precise localization of human keypoints, encouraging the encoder to learn fine-grained, pose-aware representations. To address this limitation, we introduce a pose-specific Vision Transformer (Goel et al., 2023) pretrained on human pose estimation into the visual pipeline, as illustrated in Figure 1.

Let $f_a$ denote the CLIP visual encoder and $f_b$ the pose-aware encoder. Given an input image $x$, we extract two sets of visual embeddings: $v_a = f_a(x) \in \mathbb{R}^{L_v \times d_a}$ and $v_b = f_b(x) \in \mathbb{R}^{L_v \times d_b}$, where $L_v$ is the number of visual tokens, and $d_a$, $d_b$ are the respective embedding dimensions. While UniPose (Li et al., 2025c) directly concatenates $v_a$ and $v_b$ along the channel dimension and applies a single linear projection $W \in \mathbb{R}^{(d_a + d_b) \times d}$ this design can lead to patch-level misalignment due to the differing preprocessing pipelines of the two encoders, potentially degrading performance.

To preserve the individual strengths of each visual encoder, we propose to project $v_a$ and $v_b$ individually using two separate linear layers:

$$v_a' = W_a v_a, \quad v_b' = W_b v_b, \quad W_a \in \mathbb{R}^{d_a \times d}, \quad W_b \in \mathbb{R}^{d_b \times d} \tag{11}$$

The transformed features $v_a'$ and $v_b'$ are then used in a token-level fusion with language features during joint training. This design maintains the representational integrity of both visual encoders while aligning their output with the language model's embedding space.

## B THEORETICAL DERIVATION OF THE HYGRPO OBJECTIVE

Our goal is to optimize a hybrid policy $\pi_\theta(a, p|q)$, where $a$ is a discrete action (e.g., text sequence), and $p$ is continuous action (e.g., 3D human pose), both conditioned on the input $q$. We assume the policy factorizes as:

$$\pi_\theta(a, p|q) = \pi_d(a|q) \cdot \pi_c(p|q, a), \tag{12}$$

where $\pi_d$ is the discrete policy and $\pi_c$ is the continuous policy conditioned on the discrete output. To simplify the derivation, we temporarily exclude the clipping and KL regularization terms from the GRPO (Shao et al., 2024) objective. These components are included in the final training objective but are omitted here for clarity. We begin with the simplified form of the GRPO objective :

$$\mathbb{E}_{q \sim \mathcal{D}, \{a_i, p_i\}_{i=1}^G \sim \pi_\theta(\cdot|q)} \left[ \frac{1}{G} \sum_{i=1}^G r_i(\theta) \hat{A}_i \right]. \tag{13}$$

Here, $r_i(\theta)$ is the importance weight of the $i$-th sampled output, computed as the ratio between the current policy and the reference policy:

$$r_i(\theta) = \frac{\pi_\theta(a_i, p_i|q)}{\pi_{\text{ref}}(a_i, p_i|q)} = \underbrace{\frac{\pi_\theta(a_i|q)}{\pi_{\text{ref}}(a_i|q)}}_{r_d(a_i|q)} \cdot \underbrace{\frac{\pi_\theta(p_i|q, a_i)}{\pi_{\text{ref}}(p_i|q, a_i)}}_{r_c(p_i|q, a_i)}. \tag{14}$$

To effectively train the hybrid policy, we decompose the surrogate loss into discrete and continuous components. This is motivated by the nature of our task design, where the rewards are defined separately for the discrete and continuous outputs: textual rewards $R_d(q, a)$ evaluate the semantic correctness of the generated answer $a$. pose rewards $R_c(q, a, p)$ measures the plausibility and relevance of the generated pose $p$ conditioned on both the question and answer.

Accordingly, we decompose the advantage estimate into discrete and continuous components:

$$\hat{A}(q, a, p) = \underbrace{\hat{F}(q, a)}_{\text{discrete advantages}} + \underbrace{\hat{\Delta}(q, a, p)}_{\text{continuous advantages}} . \tag{15}$$

This decomposition does not rely on an additive assumption over a shared reward function. Instead, it reflects the fact that the discrete and continuous components are supervised by independent reward signals tailored to their modalities. Accordingly, we compute two independent advantages from these separate rewards, using per-sample normalization within the candidate set:

$$\hat{F}(q, a_i) = \frac{R_d^{(i)} - \text{mean}(\{R_d\}_{i=1}^G)}{\text{std}(\{R_d\}_{i=1}^G)} \qquad \hat{\Delta}_i(q, a_i, p_i) = \frac{R_c^{(i)} - \text{mean}(\{R_c\}_{i=1}^G)}{\text{std}(\{R_c\}_{i=1}^G)}. \tag{16}$$

We now substitute this decomposition into the GRPO objective. To facilitate this, we first move the expectation over the continuous action into an inner term:

$$\mathbb{E}_{q \sim \mathcal{D}, \{a_i\}_{i=1}^G \sim \pi_\theta(\cdot|q)} \left[ \frac{1}{G} \sum_{i=1}^G r_d(a_i|q) \underbrace{\mathbb{E}_{p_i \sim \pi_\theta(p|q, a_i)}[r_c(p_i|q, a_i)\hat{A}_i(q, a_i, p_i)]}_{=:G(q, a_i)} \right], \tag{17}$$

We then analyze the inner term $G(q, a_i)$ by substituting the advantage decomposition:

$$
\begin{aligned}
G(q, a_i) &= \mathbb{E}_{p_i \sim \pi_\theta(p|q, a_i)}[r_c(p_i|q, a_i)\hat{F}_i(q, a_i) + r_c(p_i|q, a_i)\hat{\Delta}_i(q, a_i, p_i)] \\
&= \hat{F}(q, a_i) \underbrace{\mathbb{E}_{p_i \sim \pi_\theta(p|q, a_i)}[r_c(q, a_i, p_i)]}_{=1} + \mathbb{E}_{p_i \sim \pi_\theta(p|q, a_i)}[r_c(q, a_i, p_i)\hat{\Delta}_i(q, a_i, p_i)] \\
&= \hat{F}_i(q, a_i) + \mathbb{E}_{p_i \sim \pi_\theta(p|q, a_i)}[r_c(q, a_i, p_i)\hat{\Delta}_i(q, a_i, p_i)].
\end{aligned}
\tag{18}
$$

Substituting $G(q, a_i)$ back into the outer expectation, we arrive at a natural decomposition into two components:

$$\underbrace{\mathbb{E}_{q \sim \mathcal{D}, \{a_i\}_{i=1}^G \sim \pi_\theta(\cdot|q)}[r_d(a_i|q)\hat{F}_i(q, a_i)]}_{\mathcal{J}_d} + \underbrace{\mathbb{E}_{q \sim \mathcal{D}, \{a_i, p_i\}_{i=1}^G \sim \pi_\theta(\cdot|q)}[r_d(q, a_i)r_c(q, a_i, p_i)\hat{\Delta}_i(q, a_i, p_i)]}_{\mathcal{J}_c}. \tag{19}$$

Although the discrete importance weight $r_d(a|q)$ and the continuous policy $\pi_\theta(p|q, a)$ share a common embedding space and are thus implicitly coupled through shared parameters, $r_d(a|q)$ does not directly depend on the parameters of the continuous branch. In practice, when generating valid continuous 3D poses, the discrete answers $q$ are highly templated. Thus, $r_d(a|q)$ can be treated as a constant with respect to the optimization of the continuous component. Therefore, the continuous policy gradient is proportional to:

$$\nabla_\theta J_c \propto \nabla_\theta \mathbb{E}_{q \sim \mathcal{D}, \{a_i, p_i\}_{i=1}^G \sim \pi_\theta(\cdot|q)}[r_c(q, a_i, p_i)\hat{\Delta}_i(q, a_i, p_i)]. \tag{20}$$

Based on the decomposed gradient structure, we apply PPO-style (Schulman et al., 2017) clipping separately to the discrete and continuous components to stabilize training:

$$
\begin{aligned}
\mathcal{J}_{\text{HyGRPO}} = \mathbb{E}_{(q, a, p) \sim \mathcal{D}, \{a_i, p_i\}_{i=1}^G \sim \pi_\theta(\cdot|q)} \Bigg[ &\frac{1}{G} \sum_{i=1}^G \Big( \min(r_d \hat{F}_i, \text{clip}(r_d, 1-\epsilon, 1+\epsilon)\hat{F}_i) \Big) \\
&+ \frac{1}{V} \sum_{i=1}^V \Big( \min(r_c \hat{\Delta}_i, \text{clip}(r_c, 1-\epsilon, 1+\epsilon)\hat{\Delta}_i) \Big) - \beta D_{\text{KL}}(\pi_\theta \| \pi_{\text{ref}}) \Bigg].
\end{aligned}
\tag{21}
$$

Table 4: Hyperparameter settings of pose-specific MLLM pretraining and reinforcement fine-tuning.

| Hyperparameters | Pretraining | Reinforcement fine-tuning |
|---|---|---|
| Batch Size | 80 | 16 |
| Learning Rate | 3e-4 | 1e-6 |
| Training Steps | 10000 | 1000 |
| Optimizer | AdamW | AdamW |
| Adam $\beta$ | (0.9, 0.95) | (0.9, 0.95) |
| LR Schedule | Cosine | Cosine |
| Computing Resources | NVIDIA A100 (40GB) | NVIDIA A800 (80GB) |

where $G$ is the total number of sampled candidates per input, and $V \leq G$ is the number of candidates with valid continuous outputs. This objective enables separate, stable, and reward-aligned optimization of discrete and continuous policy branches within a unified reinforcement learning framework.

## C  ALGORITHM

---

**Algorithm 1 Hybrid Action Space Group Relative Policy Optimization**

---

1: **Input:** Initial policy model $\pi_{\text{init}}$ (a pretrained pose-specifc MLLM); reward models
      $R_\varphi$; task dataset $\mathcal{D}$; hyperparameters $\epsilon$
2: **Output:** policy model $\pi_\theta$

3: Initialize $\pi_\theta \leftarrow \pi_{\text{init}}, \pi_{\text{ref}} \leftarrow \pi_{\text{init}}$
4: **for** iteration $= 1, \ldots, N$ **do**
5:     Sample a batch $\mathcal{D}_b$ from $\mathcal{D}$
6:     Generate $G$ outputs $\{a_i, p_i\}_{i=1}^G \sim \pi_\theta(\cdot \mid q)$ for each question $q \in \mathcal{D}_b$
7:     Compute rewards $\{\mathcal{R}_i\}_{i=1}^G$ for each $(a_i, p_i)$ by running $\mathcal{R}_\varphi$
8:     Compute $\hat{A}_i$ for $(a_i, p_i)$ via group relative advantage estimation
9:     Update $\pi_\theta$ by maximizing HyGRPO objective Eq. 6
10: **end for**

---

## D  EXPERIMENTAL DETAILS

The detailed hyperparameter settings for both Pose-specific MLLM pretraining and reinforcement fine-tuning are provided in Table 4. In the pretraining stage, we focus on adapting the base LLaVA (Liu et al., 2023) model to 3D pose tasks, while the reinforcement fine-tuning stage further optimizes the policy behavior.

## E  MORE QUALITATIVE RESULTS: TEXT-TO-POSE

In Figure 7, we present qualitative results of Pose-RFT applied to human-written prompts from PoseScript (Delmas et al., 2022).

## F  MORE QUALITATIVE RESULTS: VIDEO-TO-POSE

In Figure 8, we present qualitative results of Pose-RFT applied to in-the-wild videos. As a frame-based model, Pose-RFT processes each frame independently, without the integration of any temporal smoothing or post-processing module.

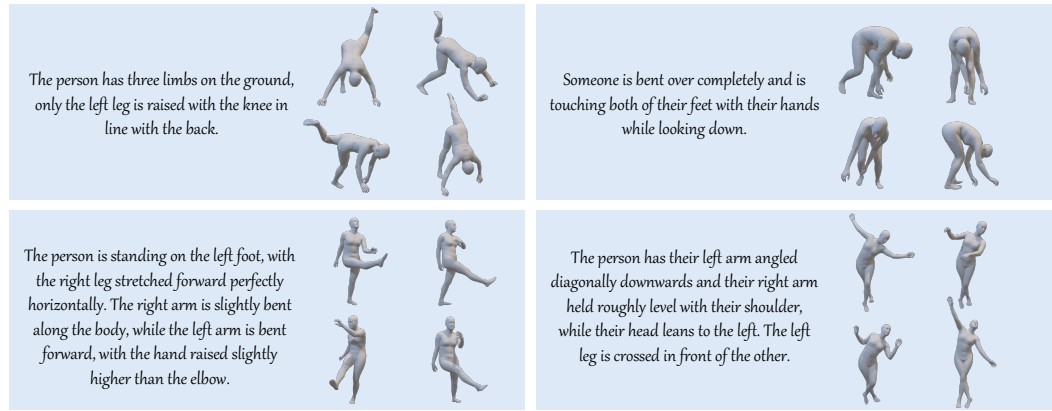

Figure 7: Pose-RFT results on human-written prompts from PoseScript (Delmas et al., 2022).

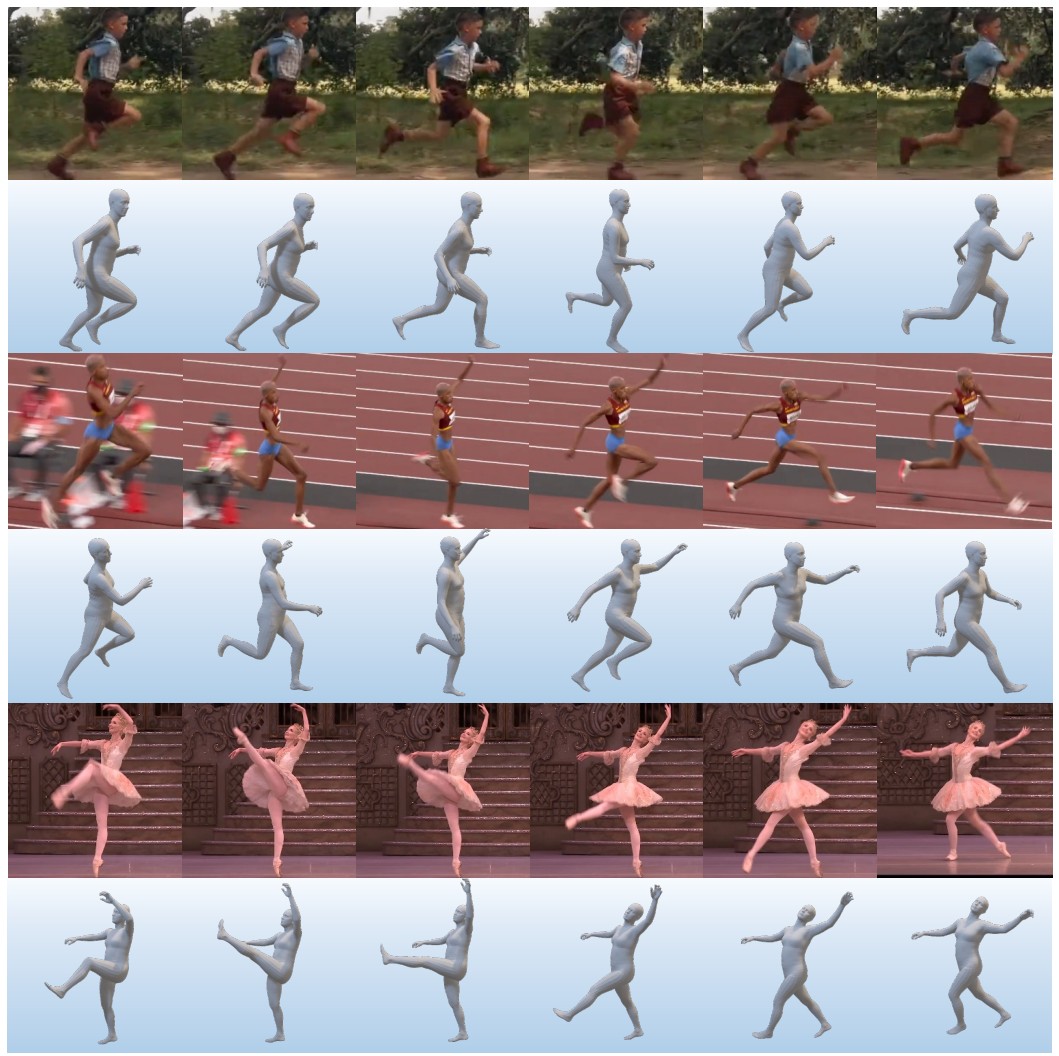

Figure 8: Pose-RFT results on in-the-wild videos.

Table 5: **Ablation study on reward components.** We report the performance impact of removing individual rewards during RL fine-tuning.

| Method | Image-to-Pose (3DPW) | | Text-to-Pose (PoseScript-H2) | |
|---|---|---|---|---|
| | MPJPE ↓ | PA-MPJPE ↓ | $R^{T2P}$ ↑ | $R^{P2T}$ ↑ |
| Baseline (no RL) | 91.4 | 59.2 | 37.4 | 42.0 |
| w/o Joint Location Reward | 108.7 | 73.5 | 55.9 | 60.3 |
| w/o Semantic Alignment Reward | 84.0 | 51.3 | 35.2 | 40.8 |
| w/o Format Reward | 131.9 | 80.6 | 28.3 | 34.4 |
| w/o Text Emb. Sim. Reward | 89.8 | 58.7 | 42.3 | 46.5 |
| **Pose-RFT (Full Model)** | 85.9 | 51.6 | 53.6 | 57.6 |

Table 6: Cross-Model Evaluation using PoseEmbroider Retrieval Model.

| | Text-to-Pose ($R^{T2P}$) ↑ | | | Pose-to-Text ($R^{P2T}$) ↑ | | |
|---|---|---|---|---|---|---|
| Method | R@5 | R@10 | R@20 | R@5 | R@10 | R@20 |
| Baseline | 47.7 | 78.3 | 86.1 | 46.2 | 80.3 | 88.7 |
| Pose-RFT (Ours) | 55.2 | 85.9 | 92.1 | 51.6 | 85.2 | 92.0 |

# G  ABLATION ON REWARD COMPONENT

To validate the efficacy of our proposed reward design, we performed an ablation study by systematically excluding individual reward terms during RL fine-tuning. Specifically, we zero-masked the target reward signal while maintaining the multi-task training pipeline to strictly isolate the contribution of each component. As shown in Table 5, the removal of any single reward leads to performance degradation in its corresponding domain. For instance, excluding the Joint Location Reward significantly impairs geometric accuracy (PA-MPJPE ↑ 21.9mm), while removing the Semantic Alignment Reward deteriorates text-pose consistency. Notably, the Format Reward proves critical for overall stability; its absence results in catastrophic failure across both modalities, confirming that structured output constraints are a prerequisite for effective optimization.

# H  CROSS-RETRIEVER EVALUATION FOR TEXT-TO-POSE

To further validate the generalization capability of our approach and ensure that the learned semantic alignment is intrinsic rather than specific to the reward model's feature space, we conducted a cross-retriever evaluation using an independent pose retrieval framework. Specifically, we employed the retrieval model from PoseEmbroider (Delmas et al., 2024) as an external evaluator in Table 6.

While PoseEmbroider shares a similar encoder architecture (VPoser + Transformer) with our reward model (PoseScript), it serves as a robust out-of-distribution test due to two fundamental differences: Data Distribution Shift: PoseEmbroider was trained on BEDLAM-Script (Black et al., 2023), a dataset derived from high-fidelity synthetic avatars. This represents a significant domain shift from the AMASS (Mahmood et al., 2019) data used to train our reward model. Distinct Training Framework: Unlike the joint embedding approach of PoseScript, PoseEmbroider utilizes a distinct multi-modal embroidery framework with uni-modal contrastive objectives.

# I  LIMITATIONS

While our method represents a promising step toward reinforcement learning-based 3D human pose generation, it has several limitations. First, its effectiveness is inherently constrained by the quality of the reward functions. Designing reliable and semantically meaningful reward signals for pose generation remains a challenging problem, especially when capturing nuanced human preferences

such as plausibility, naturalness, or contextual relevance. Inaccurate or incomplete reward feedback may misguide policy optimization, leading to suboptimal or unnatural poses.

Second, our framework relies on sampling multiple candidate responses per input to perform group-wise reward normalization. Although this design improves training stability in hybrid action spaces, it introduces non-negligible computational overhead, which may limit scalability when applied to larger models or datasets.

## J    THE USAGE OF LARGE LANGUAGE MODELS (LLMS)

We used a Large Language Model (LLM) as a general-purpose writing assistant to improve the clarity, grammar, and style of the manuscript. The LLM provided suggestions for sentence phrasing and readability, but all content, ideas, and scientific claims in this paper are the sole responsibility of the authors.

