# OpenReview forum: "Pose-RFT: Aligning MLLMs for 3D Pose Generation via Hybrid Action Reinforcement Fine-Tuning"
_ICLR.cc/2026/Conference — ICLR 2026 Poster_

### Official Review · Reviewer_6Rpp · 2025-10-30

**Soundness:** 3
**Presentation:** 3
**Contribution:** 3
**Rating:** 6
**Confidence:** 3

**Summary:**

The paper addresses the alignment gap in text/image-to-pose generation, where the one-to-many nature of 3D poses makes supervised fine-tuning (SFT) suboptimal. The authors propose Pose-RFT, framing the task as hybrid-action reinforcement learning and introducing HyGRPO for joint optimization over discrete (text) and continuous (pose) spaces. HyGRPO normalizes group-wise rewards and decomposes the advantage into discrete (language) and continuous (pose) components. Four task-specific reward functions are designed: spatial, semantic, format, and text-similarity.

**Strengths:**

*Originality:* The motivation is clear and well-grounded. Applying reinforcement fine-tuning to 3D pose generation is novel and well-justified given the one-to-many mapping nature.

*Quality:* Extensive experiments across multiple datasets (3DPW, H36M, PoseScript) demonstrate SOTA performance against MLLM baselines. Ablations are comprehensive.

*Clarity:* Writing and figures are clear and intuitive.

*Significance:* Introducing RFT into multimodal 3D pose generation is timely and impactful for the community.

**Weaknesses:**

1. **Lack of ablation on reward components.**
   Although the paper emphasizes the importance of the four reward functions, it provides no analysis isolating their individual contributions. Section 4.3 only examines Pose-aware Encoder, distribution modeling, and RFT as a whole. Hence, it remains unclear which reward(s) primarily drive the improvements—semantic alignment, spatial accuracy, or the mere presence of RL signal.

2. **Reward–evaluation entanglement.**
    Some reward terms use the same embedding models as the evaluation metrics (e.g., BGE-M3, PoseScript retriever), creating potential bias or “reward hacking.” Cross-evaluator validation would strengthen claims of generality.

3. **Computational overhead.**
   HyGRPO requires sampling multiple candidates and performing group normalization. While the appendix mentions resource cost, there are no efficiency statistics or scaling curves quantifying the trade-off.

**Questions:**

1. Have the authors conducted reward ablations? If not, please clarify which components are most influential.
2. Could you evaluate under a different embedding/retrieval model to test generalization?
3. How do key hyperparameters (group size G, continuous sample count V) affect performance, efficiency, and stability?

**Details Of Ethics Concerns:**

The paper uses anonymous skeleton data and discusses potential biases and misuse.

---

> ### Author Response · Authors · 2025-11-21
> **Rebuttal to Reviewer 6Rpp - Part 1**
>
> # Summary Response
>
> We sincerely appreciate the reviewer’s constructive feedback and insightful suggestions. Below, we provide point-by-point responses.
>
> -----
>
> > **W1:** **Lack of ablation on reward components.** Although the paper emphasizes the importance of the four reward functions, it provides no analysis isolating their individual contributions. Section 4.3 only examines Pose-aware Encoder, distribution modeling, and RFT as a whole. Hence, it remains unclear which reward(s) primarily drive the improvements—semantic alignment, spatial accuracy, or the mere presence of RL signal.
> >
> > **Q1:** Have the authors conducted reward ablations? If not, please clarify which components are most influential.
>
> ### **Ablation on Reward Components**
>
> To empirically validate this design and assess the importance of each component, we conducted an ablation study where we removed one reward function at a time during RL fine-tuning. Methodologically, for batches corresponding to the removed reward, we **masked the reward signal with a constant scalar (zero)**. This approach preserves the stability of the multi-task training pipeline while strictly isolating the impact of the specific reward function. We have added this experiment to **Appendix G**.
>
> | Model                                | Image-to-Pose (3DPW MPJPE/PA-MPJPE $\downarrow$) | Text-to-Pose (PoseScript-H2 $\text{mRecall}^{T2P}$ / $\text{mRecall}^{P2T} \uparrow$ ) |
> | ------------------------------------ | ------------------------------------------------------ | ------------------------------------------------------------ |
> | Baseline (no RL)                     | 91.4 / 59.2                                            | 37.4 / 42.0                                                  |
> | w/o Joint Location Reward            | 108.7 / 73.5                                           | 55.9 / 60.3                                                  |
> | w/o Semantic Alignment Reward        | 84.0 / 51.3                                            | 35.2 / 40.8                                                  |
> | w/o Format Reward                    | 131.9 / 80.6                                           | 28.3 / 34.4                                                  |
> | w/o Text Embedding Similarity Reward | 89.8 / 58.7                                            | 42.3 / 46.5                                                  |
> | Pose-RFT (Full Model)                | 85.9 / 51.6                                            | 53.6 / 57.6                                                  |
>
> The results show that removing each reward leads to a measurable drop in the performance of the corresponding task, confirming that each reward provides essential and non-redundant feedback signals.
>
> -----
>
> > **W2:** **Reward–evaluation entanglement.** Some reward terms use the same embedding models as the evaluation metrics (e.g., BGE-M3, PoseScript retriever), creating potential bias or “reward hacking.” Cross-evaluator validation would strengthen claims of generality.
> >
> > **Q2**: Could you evaluate under a different embedding/retrieval model to test generalization??
>
> ### **Cross-Retriever Evaluation**
>
> We thank the reviewer for identifying this critical methodological concern regarding the potential for "reward hacking" or "test-taking". To prove that our model has achieved true semantic alignment rather than merely overfitting the reward model's feature space, we conducted a Cross-Retriever Evaluation. We have added this experiment to **Appendix H**.
>
> We employed the **retrieval model from PoseEmbroider [1]** as an independent evaluator. While PoseEmbroider shares a similar encoder architecture (VPoser + Transformer) with our reward model (PoseScript), it differs fundamentally in two key aspects:
>
> - Data Distribution: It was trained on BEDLAM-Script —a different dataset derived from high-fidelity synthetic avatars.
> - Training Framework: It utilizes a distinct multi-modal embroider framework with uni-modal contrastive objectives.
>
> |                  | $R^{T2P} \uparrow$ | $R^{P2T} \uparrow$ |
> | ---------------- | ------------------ | ------------------ |
> | Baseline       | 47.7 / 78.3 / 86.1 | 46.2 / 80.3 / 88.7 |
> | Pose-RFT (Ours) | 55.2 / 85.9 / 92.1 | 51.6 / 85.2 / 92.0 |
>
> *Note: We strictly follow PoseEmbroider [1] fitting procedure to convert our generated body model to SMPL-X format before feature extraction, ensuring a rigorous and fair comparison.*
>
>
>
> [1] PoseEmbroider: Towards a 3D, Visual,  Semantic-aware Human Pose Representation; ECCV 2024

---

> ### Author Response · Authors · 2025-11-21
> **Rebuttal to Reviewer 6Rpp - Part 2**
>
> -----
>
> > **W3**: **Computational overhead.** HyGRPO requires sampling multiple candidates and performing group normalization. While the appendix mentions resource cost, there are no efficiency statistics or scaling curves quantifying the trade-off.
>
> ### **Efficiency Analysis:**
>
> We thank the reviewer for this important question. We provide a detailed comparison of training and inference efficiency between SFT and our Pose-RFT. All experiments were conducted on 4 NVIDIA A100 GPUs.
>
> **Inference Efficiency:** Our RFT approach introduces **zero computational overhead** at inference time. The final model (Pose-RFT) uses the same network architecture as the SFT baseline, just with updated weights. Therefore, the inference speed (e.g., latency and throughput) is identical.
>
> **Training Efficiency:** We compare the computational cost of the SFT stage (5,000 steps) versus our RFT stage (1,000 steps).
>
> - **SFT Stage:** Trained for 5,000 steps with a batch size of 80.
>
> - **RFT Stage:** Trained for only 1,000 steps with a batch size of 16 and a group size of 8.
>
> | Training Stage | Total Training Time (h) | Time Per Sample (s) |
> | -------------- | ----------------------- | ------------------- |
> | **SFT**        | 16.21                   | 0.1459              |
> | **RFT**        | 3.86                    | 0.8688              |
>
> Critically, we wish to clarify that SFT and RFT are complementary stages, not competing alternatives. Following the standard training recipe for modern MLLMs (Pretraining $\to$ SFT $\to$ RFT), our approach adapts this paradigm to the domain of 3D pose generation.
>
> -----
>
> > **Q3**: How do key hyperparameters (group size G, continuous sample count V) affect performance, efficiency, and stability?
>
> ### **Impact of Hyperparameters**
>
> We thank the reviewer for this insightful question regarding the learning dynamics of HyGRPO.
>
> **Clarification on $V$:** First, we clarify that the continuous sample count $V$ is not a fixed hyperparameter but an **automatic subset** of $G$. It represents the number of candidates within a group of size $G$ that successfully generate valid pose formats. Therefore, the governing hyperparameter is the Group Size $G$.
>
> **Ablation Study on Group Size $G$:** We conducted an ablation study varying $G \in$ {4, 8, 16} to analyze the trade-offs between Performance, Efficiency, and Stability.
>
> - Performance: Measured by PA-MPJPE on 3DPW and mRecall on PoseScript-H2.
> - Efficiency: Measured by the normalized Training Time.
> - Stability: Quantified by the Average Clip Ratio. A high clip ratio indicates frequent trust region violations due to high-variance advantage estimates (instability).
>
> | Group Size ($G$) | Performance                                  | Efficiency                   | Stability                      |
> | ---------------- | -------------------------------------------- | ---------------------------- | ------------------------------ |
> |                  | (PA-MPJPE $\downarrow$ / mRecall $\uparrow$) | (Training Time $\downarrow$) | (Avg. Clip Ratio $\downarrow$) |
> | $G$ = 4          | 55.1 mm / 46.8%                              | 1.0x                         | 12.8%                          |
> | $G$ = 8 (Ours)   | 51.6 mm / 55.6%                              | 1.4x                         | 3.5%                           |
> | $G$ = 16         | 51.9 mm / 56.0%                              | 2.3x                         | 3.2%                           |
>
> The results show that small groups suffer from high instability (12.8% clip ratio), while $G=8$ strikes the optimal balance, reducing the clip ratio to 3.5% with minimal computational overhead compared to $G=16$.

---

### Official Review · Reviewer_kH8o · 2025-10-31

**Soundness:** 3
**Presentation:** 3
**Contribution:** 3
**Rating:** 6
**Confidence:** 3

**Summary:**

This paper aims to address the alignment gap problem in current 3D human pose generation methods. The authors point out that existing methods primarily rely on Supervised Fine-Tuning, learning a deterministic mapping from input to a single ground truth pose through regression and other methods. However, 3D pose generation tasks (whether text-to-pose or image-to-pose) inherently possess a "one-to-many" ambiguity, which the SFT paradigm struggles to handle.

To address this issue, the paper proposes the Pose-RFT framework, using reinforcement learning fine-tuning. The core technical challenge lies in handling a hybrid action space, requiring simultaneous optimization of discrete text outputs and continuous 3D pose parameters. To this end, the authors design a novel online reinforcement learning algorithm called **HyGRPO** (Hybrid Action Space Group Relative Policy Optimization). This algorithm achieves stable optimization of the hybrid policy by grouping and normalizing the rewards of a sampled set of candidate outputs.

Furthermore, the paper designs a set of task-specific reward functions to guide spatial alignment for image-to-pose tasks and semantic consistency for text-to-pose tasks, respectively. Extensive experiments on multiple 3D pose generation benchmarks demonstrate that Pose-RFT achieves significant performance improvements over existing pose-specific MLLMs.

**Strengths:**

- In the context of MLLM, the authors define pattern collapse in generative models as "alignment gap”. Furthermore, the authors demonstrate why a shift from supervised imitation to a reward-driven optimization paradigm is necessary.
- The authors demonstrate that RL can not only be used to align a model's internal understanding, but also to directly guide the model to generate complex, structured non-textual data.
- The authors propose the HyGRPO algorithm to provide independent and targeted feedback signals for different output heads of the model, which is a simple and logically consistent approach.

**Weaknesses:**

- Pose-RFT achieved PA-MPJPE values of 44.5 mm and 85.9 mm on Human 3.6M and 3DPW, respectively. The paper mentions a "performance gap" compared to traditional expert models, but quickly shifts its focus to its state-of-the-art (SOTA) performance on the RPE task. However, this performance gap is not insignificant, but rather quite substantial. Current SOTA expert models already have PA-MPJPE values far below this value on Human 3.6M, for example, 29.1 in ([CVPR 2023 paper](https://openaccess.thecvf.com/content/CVPR2023/html/Fang_Learning_Analytical_Posterior_Probability_for_Human_Mesh_Recovery_CVPR_2023_paper.html)) and 29.4 in ([Springer link](https://link.springer.com/chapter/10.1007/978-3-031-72640-8_27)). This may be due to differences in assessment protocols, which the authors need to clarify, but it undoubtedly underestimates the true gap. The authors should consider repositioning the paper's contribution as "bringing new interactive and inference capabilities to pose estimation," rather than merely competing on traditional metrics.

- In the text-to-pose task, the authors used a pre-trained text-pose retrieval model to define the semantic alignment reward. However, in the evaluation phase, they also used the same retrieval model (derived from the PoseScript paper) to calculate the Recall@K metric. This design leads to a methodological loop: the RFT process directly optimizes the model to generate poses that score high on the similarity metric of the retrieval model, and then uses the same metric to evaluate its performance. This can lead to overfitting to the evaluation metric, where the generated poses may not appear more reasonable or accurate to human observers, but are simply better at matching the feature space of the evaluation tool. This is a serious limitation that undermines the credibility of the results for the text-to-pose generation task.

- HyGRPO appears to be an adaptation of the GRPO algorithm in a hybrid action space. While this application is novel, its core algorithmic components likely have stronger precedents in the broader hybrid RL literature than the paper acknowledges. The paper should more clearly compare its algorithm with these existing works to clarify its exact novelty.

- The paper proposes four independent reward functions, but does not discuss the weight balance and potential conflicts among them.

**Questions:**

1. On standard benchmarks such as Human3.6M and 3DPW, your approach exhibits a significant performance gap compared to domain expert models. Could you elaborate on the reasons for this gap and articulate it as a fundamental trade-off between using a general-purpose MLLM architecture and a specialized architecture?

2. Evaluation of text-to-gesture tasks relies on the same retrieval model that provides the semantic reward signal, which can lead to a "test-taking" problem. How do you view this potential methodological limitation? Have you considered employing alternative evaluation strategies?

---

> ### Author Response · Authors · 2025-11-21
> **Rebuttal to Reviewer kH8o - Part 1**
>
> # Summary Response
>
> We sincerely appreciate the reviewer’s constructive feedback and insightful suggestions. Below, we provide point-by-point responses.
>
> -----
>
> > **Response to W1 & Q1:**
>
> ### **Performance Gap & Repositioning**
>
> We thank the reviewer for this constructive suggestion. We fully agree that the core contribution of Pose-RFT lies in "bringing new interactive and inference capabilities to pose estimation" rather than merely competing on traditional reconstruction metrics.
>
> Accordingly, we have revised our claims to emphasize the synergy between specialist precision and generalist reasoning. We demonstrate that our architectural design enables an **Integrated Pose System**, where the high-precision specialist and the reasoning-capable MLLM are not mutually exclusive but complementary components sharing a common visual backbone.
>
> **1. New Capability: Reasoning Pose Estimation (RPE):**
>
> As noted by the reviewer, the unique value of MLLM-based frameworks is their ability to process multimodal context, enabling capabilities that traditional specialist models cannot achieve:
>
> - **Referring Pose Estimation in Crowds**: When multiple individuals are present, visual cues alone often fail to determine *which* person to estimate. Our model leverages natural language (e.g., *"the child holding the red balloon"*) to locate and estimate the pose of the specific individual of interest.
> - **Robust pose estimation under occlusion**:  Language can serve as a complementary modality that helps the model infer plausible 3D body configurations even when parts of the body are not visible.
>
> As shown in Table 1 of our paper, Pose-RFT achieves **SOTA performance on the RPE benchmark**, validating this unique reasoning advantage
>
> **2. Bridging the Performance Gap: An Integrated Pose System**
>
> To address the performance gap on standard benchmarks while retaining reasoning capabilities, we analyzed the model's performance across four evolution stages. We propose that Pose-RFT should be viewed as an integrated system where the visual specialist is an inherent component (the Pose-Aware Encoder) that can be leveraged directly **when textual reasoning is not required**.
>
> To rigorously analyze the source of the performance gap and justify our architectural choices, we conducted a controlled **decomposition experiment** across four evolution stages.
>
> - Stage I: Visual Specialist. We first trained a lightweight regression head directly on top of our frozen Pose-Aware Encoder, bypassing the LLM entirely. Trained solely on Image-to-Pose data, this "Specialist" achieves a PA-MPJPE of 35.6 mm , which is competitive with traditional SOTA methods. This confirms that our visual backbone captures sufficient spatial details.
> - Stage II: Single-Task Alignment. Next, we integrated this encoder into the MLLM and applied Pose-RFT, but trained only on Image-to-Pose data. This performance drop represents the "Alignment Cost"—the primary cost when mapping visual features into a language-aligned LLM feature space.
> - Stage III: Full Multi-Task Generalist. Then, our full Pose-RFT model is trained on mixed tasks (Image-to-Pose, Text-to-Pose, VQA). While the PA-MPJPE settles at 44.5 mm (a slight trade-off due to task interference), this stage unlocks the critical Reasoning Pose Estimation (RPE) capability.
> - **Stage IV: Integrated Pose System.** Finally, since the Visual Specialist (Stage I) and the Generalist (Stage III) share the same frozen visual backbone, our framework can inherently support both. In practical applications, the system can adaptively leverage the Specialist Head when maximal precision is required for standard images, or switch to the Generalist Head when textual reasoning and interaction are needed. This design effectively bridges the performance gap without sacrificing the unique capabilities of MLLMs.
>
> | Stage | Model Varariant                  | Achitecture                             | Training Scope            | Human3.6M (MPJPE / PA-MPJPE $\downarrow$) | Reasoning Pose Estimation (MPJPE / PA-MPJPE $\downarrow$) |
> | :---: | -------------------------------- | --------------------------------------- | ------------------------- | ----------------------------------------- | ---------------------------- |
> |   I   | Visual Specialist                | Pose-Aware Enc. Only  (Regression Head) | Image-to-Pose Only        | 46.8 / 35.6                               | 229.2 / 107.3                |
> |  II   | Pose-RFT (Single-Task Alignment) | MLLM                                    | Image-to-Pose Only        | 61.8 / 44.2                               | 234.9 / 109.7                |
> |  III  | Pose-RFT (Full-Task Generalist)  | MLLM                                    | Multi-Task | 63.0 / 44.5                               | 198.6 / 87.0                 |
> |  IV   | Integrated Pose System    | Pose-Aware Enc. + MLLM                  | Multi-Task | **46.8 / 35.6**                           | **198.6 / 87.0**             |

---

> ### Author Response · Authors · 2025-11-21
> **Rebuttal to Reviewer kH8o - Part 2**
>
> -----
> > **Response to W2:**
>
> ### **Cross-Retriever Evaluation**
>
> We thank the reviewer for identifying this critical methodological concern regarding the potential for "reward hacking" or "test-taking". To prove that our model has achieved true semantic alignment rather than merely overfitting the reward model's feature space, we conducted a Cross-Retriever Evaluation. We have added this experiment to **Appendix H**.
>
> We employed the **retrieval model from PoseEmbroider [1]** as an independent evaluator. While PoseEmbroider shares a similar encoder architecture (VPoser + Transformer) with our reward model (PoseScript), it differs fundamentally in two key aspects:
>
> - Data Distribution: It was trained on BEDLAM-Script —a different dataset derived from high-fidelity synthetic avatars.
> - Training Framework: It utilizes a distinct multi-modal embroider framework with uni-modal contrastive objectives.
>
> |                  | $R^{T2P} \uparrow$ | $R^{P2T} \uparrow$ |
> | ---------------- | ------------------ | ------------------ |
> | Baseline       | 47.7 / 78.3 / 86.1 | 46.2 / 80.3 / 88.7 |
> | Pose-RFT (Ours) | 55.2 / 85.9 / 92.1 | 51.6 / 85.2 / 92.0 |
>
> *Note: We strictly follow PoseEmbroider [1] fitting procedure to convert our generated body model to SMPL-X format before feature extraction, ensuring a rigorous and fair comparison.*
>
>
>
> [1] PoseEmbroider: Towards a 3D, Visual,  Semantic-aware Human Pose Representation; ECCV 2024
>
> -----
>
> > **Response to W3:**
>
> ### **Novelty & Comparison with Hybrid RL Literature**
>
> We appreciate the reviewer's deep insight into the Hybrid RL literature. We acknowledge that our work builds upon the foundational concepts of this field. We clarify the architectural and algorithmic distinctions below:
>
> **1. Architectural Choice: Parallel Heads (vs. HyAR)**
>
> - HyAR addresses hybrid action spaces by compressing discrete and continuous actions into a unified latent continuous representation via a conditional VAE, training a single latent policy.
> - In MLLMs, the shared feature space is already semantically rich and aligned. we adopt a Parallel Sub-Actor architecture, utilizing two specialized heads rooted in the shared LLM backbone: a Discrete Head for semantic reasoning (text) and a Continuous Head for spatial regression. This ensures that each output modality is handled by the most appropriate architectural component.
>
> **2. Optimization Paradigm: The "Critic-Free" Evolution (vs. H-PPO)**
>
> - While H-PPO successfully uses parallel actors, it relies heavily on a centralized **Critic Network** (State-Value Function) to estimate advantages. Training a separate Value Network for a 7B-parameter MLLM is computationally prohibitive and prone to instability.
> - HyGRPO represents a **Critic-Free** evolution of the H-PPO framework. Instead of training a Value Network, we introduce **Group Relative Advantage Estimation** to the hybrid setting. We compute advantages by comparing a group of sampled hybrid outputs $\{a_i, p_i\}_{i=1}^G$ against each other. This allows us to optimize the decoupled discrete and continuous heads efficiently without the bottleneck of value function approximation.
>
> **3. Infeasibility of Value-Based Baselines (P-DQN, MP-DQN)**
>
> - While classical algorithms like P-DQN [4] and MP-DQN [5] are fundamental to Hybrid RL, they rely on estimating $Q(s, k, x_k)$ for all discrete actions $k$. In MLLMs, the discrete action space is the vocabulary size ($|\mathcal{V}| \approx 32K \text{-} 100K$). MP-DQN’s requirement to iterate or evaluate Q-values for every possible token is computationally intractable for Large Language Models. HyGRPO adopts a policy gradient approach to bypass this bottleneck.
>
> [2] Hyar: Addressing discrete-continuous action reinforcement learning via hybrid action representation; ICLR 2022
>
> [3] H-PPO: Hybrid Actor-Critic Reinforcement Learning in Parameterized Action Space; IJCAI 2019
>
> [4] P-DQN: Parametrized deep q-networks learning: Reinforcement learning with discrete-continuous hybrid action space; arXiv:1810.06394, 2018
>
> [5] MP-DQN: Multi-Pass Q-Networks for Deep Reinforcement Learning with Parameterised Action Spaces; arXiv:1905.04388, 2019

---

> ### Author Response · Authors · 2025-11-21
> **Rebuttal to Reviewer kH8o - Part 3**
>
> -----
>
> > **Response to W4:**
>
> ### **Reward Sensitivity and Ablation Study:**
>
> We thank the reviewer for this valuable suggestion. To address the concern regarding reward sensitivity, we first clarify our design philosophy: our framework utilizes **task-specific rewards**, where each reward function is formulated to guide the optimization of a distinct type of output.
>
> To empirically validate this design and assess the importance of each component, we conducted an ablation study where we removed one reward function at a time during RL fine-tuning. Methodologically, for batches corresponding to the removed reward, we **masked the reward signal with a constant scalar (zero)**. This approach preserves the stability of the multi-task training pipeline while strictly isolating the impact of the specific reward function. We have added this experiment to **Appendix G**.
>
> | Model                                | Image-to-Pose (3DPW MPJPE/PA-MPJPE ↓) | Text-to-Pose (PoseScript-H2 $\text{mRecall}^{T2P}$ / $\text{mRecall}^{P2T}$ ↑) |
> | ------------------------------------ | ------------------------------------- | ------------------------------------------------------------ |
> | Baseline (no RL)                     | 91.4 / 59.2                           | 37.4 / 42.0                                                  |
> | w/o Joint Location Reward            | 108.7 / 73.5                          | 55.9 / 60.3                                                  |
> | w/o Semantic Alignment Reward        | 84.0 / 51.3                           | 35.2 / 40.8                                                  |
> | w/o Format Reward                    | 131.9 / 80.6                          | 28.3 / 34.4                                                  |
> | w/o Text Embedding Similarity Reward | 89.8 / 58.7                           | 42.3 / 46.5                                                  |
> | Pose-RFT (Full Model)                | 85.9 / 51.6                           | 53.6 / 57.6                                                  |
>
> The results show that removing each reward leads to a measurable drop in the performance of the corresponding task, confirming that each reward provides essential and non-redundant feedback signals.

---

### Official Review · Reviewer_rGEb · 2025-10-31

**Soundness:** 2
**Presentation:** 3
**Contribution:** 2
**Rating:** 4
**Confidence:** 3

**Summary:**

Pose-RFT introduces a RL fine-tuning framework for MLLM-based 3D pose generation. This framework, HyGRPO, utilizes a mixed optimization strategy to optimize both discrete tokens (from the LLM) and continuous motion (SMPL params).

**Strengths:**

Enabling LLMs to estimate / generate poses is a highly relevant task and utilizing a RL-based framework to optimize this is a reasonable strategy.

**Weaknesses:**

My major concern is that one of the main premises of this work, that Pose-RFT is required for better semantic and spatial alignment of HMR, is misleading. The statement that the “reliance on objectives like SMPL parameter regression creates a … alignment gap, … [to] … achieve the required semantic and spatial fidelity” (l016-019) is inaccurate or unclear.
Arguably, the simplified assumptions about the context (3d world, pose, camera) are the reason for “poor” performance HMR. On one hand, generative models like Score-HMR [1] have been utilized successfully and the authors should discuss this line of work. On the other hand, more accurate modeling of the context, i.e. like has been done in CameraHMR [2] solves the problem and simply egressing the SMPL parameters actually outperforms all other methods.

One of the main contributions of the paper is HyGRPO which is an optimization algorithm to optimize both discrete and continuous outputs. This is necessary as the authors note themselves, because MLLMs operate in discrete space but poses are in continuous space. However, there is a body of work that successfully learns discrete tokenization for human poses, for example TokenHMR. I wonder if the authors could leverage the TokenHMR tokenizer instead of having to jointly optimize discrete and cont’ signals - at least as a baseline.

The authors should also evaluate on EMDB [3], which provides more accurate gt poses.


[1] ScoreHMR: Score-Guided Diffusion for 3D Human Recovery; CVPR 2024

[2] CameraHMR: Aligning People with Perspective; 3DV 2025

[3] EMDB: The Electromagnetic Database of Global 3D Human Pose and Shape in the Wild; ICCV 2023

**Questions:**

How well does this method do on videos? It would be super interesting to see how stable the model predictions would be in a temporal setting.

---

> ### Author Response · Authors · 2025-11-21
> **Rebuttal to Reviewer rGEb - Part 1**
>
> > **W1:** My major concern is that one of the main premises of this work, that Pose-RFT is required for better semantic and spatial alignment of HMR, is misleading. The statement that the “reliance on objectives like SMPL parameter regression creates a … alignment gap, … [to] … achieve the required semantic and spatial fidelity” (l016-019) is inaccurate or unclear.
> >
> > On one hand, generative models like Score-HMR [1] have been utilized successfully and the authors should discuss this line of work. On the other hand, more accurate modeling of the context, i.e. like has been done in CameraHMR [2] solves the problem and simply egressing the SMPL parameters actually outperforms all other methods.
>
> ### **Motivation of Our RFT**
>
> We sincerely thank the reviewer for the highly professional and insightful summarization of the current solutions to the ambiguity problem in Human Mesh Recovery (HMR). We fully agree with your categorization that current SOTA methods tackle this via:
>
> - **Uncertainty Modeling:** (e.g., diffusion-based ScoreHMR [1]).
>
> - **Context Modeling:** (e.g., CameraHMR [2] with its perspective camera model).
>
> We will clarify our motivation and demonstrate the necessity of RFT by addressing three key points.
>
> **1. Clarification about Our Tasks**
>
> First, we must clarify that Pose-RFT is designed for a **MLLM-based, Unified Pose Generation** framework. This task setting is fundamentally different from that of pure vision-only HMR methods like ScoreHMR [1] and CameraHMR [2].
>
> | Methods                      | Tasks                                                        |
> | :--------------------------- | :----------------------------------------------------------- |
> | Pose-RFT (Ours)              | Unified Pose Generation (Image-to-Pose, Text-to-Pose, Reasoning Pose Estimation, VQA) |
> | ScoreHMR [1] / ScoreHypo [3] | Human Mesh Recovery (Image-to-Pose only)                |
>
> Therefore, our motivation for RFT is to enact a **paradigm shift** from supervised imitation (SFT) to reward-driven optimization (RFT), which is essential for resolving this ambiguity in pose-centric MLLMs.
>
> **2. RFT is Effective also for Generative Visual HMR**
>
> The reviewer astutely points out that generative models can address ambiguity. We are asked: **in this context, is RFT still required?**
>
> We argue **yes**, and we demonstrate this by first clarifying the methods mentioned:
>
> - **ScoreHMR [1]** is not an end-to-end generative backbone, but a **test-time diffusion-based optimizer** that refines the initial output.
> - A true generative HMR backbone, as seen in **ScoreHypo [3]**, is an end-to-end diffusion model that reconstructs the mesh from noise.
>
> To prove our hypothesis that **RFT is effective for even strong visual HMR backbones**, we conducted a new experiment. We used ScoreHypo (HypoNet) [3] as the strong generative base model and applied our RFT framework to it.
>
> The results on 3DPW and Human3.6M are definitive:
>
> | Methods                                        | 3DPW (MPJPE / PA-MPJPE) | Human3.6M (MPJPE / PA-MPJPE) |
> | :--------------------------------------------- | :--------------------------- | :-------------------------------- |
> | ScoreHMR [1] (HMR + Diffusion Optimizer) | 76.8 / 51.1                  | 44.7 / 29.0                       |
> | ScoreHypo (Generative HMR Backbone)      | 61.8 / 36.1                  | 37.4 / 25.3                       |
> | ScoreHypo + RFT (Ours)                         | **57.4 / 34.8**              | **36.9 / 25.0**                   |
>
> *To conduct this experiment, we initialized our policy with the publicly available pre-trained ScoreHypo (HypoNet) [3] model. We then fine-tuned this base policy using diffusion-version GRPO algorithm, guided by the task-specific reward function.*
>
> This confirms our central thesis: **RFT is not redundant**. It is a critical component that unlocks the full potential of generative models by solving the fine-grained alignment gap that SFT and generative-only objectives leave behind.
>
> **3. Synergizing with Context Modeling**
>
> Finally, we fully agree that context modeling methods like CameraHMR [2] are powerful and **orthogonal to our contribution**.
>
> Our RFT framework is not mutually exclusive with better context; it is synergistic. A method like CameraHMR [2] provides a better state representation (a policy with a stronger prior) for our RFT, which would serve as a **better starting point** for our reward-driven optimization.
>
> As described in Appendix A, we already embrace this principle by incorporating a strong pose-aware ViT to enhance our base model's visual features. While **CameraHMR [2] is not yet public**, its ViT-based architecture could be seamlessly integrated into our MLLM framework to create an even stronger base policy, which we are confident our RFT framework would further improve.

---

> ### Author Response · Authors · 2025-11-21
> **Rebuttal to Reviewer rGEb - Part 2**
>
> ------
>
> > **W2:** One of the main contributions of the paper is HyGRPO which is an optimization algorithm to optimize both discrete and continuous outputs. This is necessary as the authors note themselves, because MLLMs operate in discrete space but poses are in continuous space. However, there is a body of work that successfully learns discrete tokenization for human poses, for example TokenHMR [4]. I wonder if the authors could leverage the TokenHMR [4] tokenizer instead of having to jointly optimize discrete and cont’ signals - at least as a baseline.
>
> ### **Challenges of Discrete Pose Representation**
>
> We thank the reviewer for this highly insightful question and for suggesting this important alternative.
>
> We did consider this "discrete pose token" approach. However, integrating it into a Multimodal Large Language Model (MLLM) framework is **not a simpler baseline**. On the contrary, it introduces a cascade of **non-trivial architectural challenges**.
>
> Our work's focus was to investigate a **new learning paradigm (RFT)** that could improve performance while minimizing architectural changes to the base MLLM. We identified two fundamental challenges with the tokenization approach in an MLLM context:We identified three fundamental challenges with the tokenization approach, which solidified our decision to develop a direct hybrid-action algorithm like HyGRPO:
>
> **1. The "Vocabulary" Challenge:** TokenHMR's "pose tokens" are learned indices from a VQ-VAE codebook . These tokens are **not language** and do not exist in the MLLM's vocabulary. To integrate them, one must perform modifications to the MLLM's embedding layer and vocabulary, forcing it to manage two distinct token spaces.
>
> **2. The "Causal vs. Parallel" Architectural Conflict (The Core Problem):** This is the most critical challenge. MLLMs are **auto-regressive** and built on **causal attention** (i.e., generating the next token based on previous tokens). This is perfect for language, which is sequential.
>
> - This is fundamentally misaligned with the nature of pose. A 3D pose, which TokenHMR learns to represent as a set of discrete tokens (160 tokens in TokenHMR) , is a parallel spatial state, not a sequential one.
>
> - A causal model (like a standard LLM) cannot properly model the dependencies between all discrete pose tokens simultaneously. Forcing a sequential, auto-regressive generation for a parallel state is a suboptimal design.
>
> ----------
>
> > **W3:** The authors should also evaluate on EMDB [5], which provides more accurate gt poses.
>
> ### **Evaluation on EMDB Benchmark**
>
> We thank the reviewer for this suggestion. We agree that EMDB [5] is an important and valuable benchmark, known for its high-quality ground-truth annotations.
>
> We applied for access to the EMDB dataset at the beginning of the rebuttal period. Unfortunately, our **application approval is still pending**, and we have not yet received the data.
>
> However, we recognize the importance of this evaluation. We firmly commit to including a full evaluation of our method on the EMDB benchmark in the final camera-ready version of the paper.
>
> ------
>
> > **Q1:** How well does this method do on videos? It would be super interesting to see how stable the model predictions would be in a temporal setting.
>
> ### **Performance on Videos**
>
> To evaluate the temporal stability, we applied our frame-based model directly to video sequences for qualitative analysis. The visualization results are provided in **Appendix F** of the revised paper.
>
> ------
>
> [1] ScoreHMR: Score-Guided Diffusion for 3D Human Recovery; CVPR 2024
>
> [2] CameraHMR: Aligning People with Perspective; 3DV 2025
>
> [3] ScoreHypo: Probabilistic Human Mesh Estimation with Hypothesis Scoring; CVPR 2024
>
> [4] TokenHMR: Advancing Human Mesh Recovery with a Tokenized Pose Representation; CVPR 2024
>
> [5] EMDB: The Electromagnetic Database of Global 3D Human Pose and Shape in the Wild; ICCV 2023

---

### Official Review · Reviewer_eur6 · 2025-11-01

**Soundness:** 3
**Presentation:** 3
**Contribution:** 3
**Rating:** 8
**Confidence:** 4

**Summary:**

This paper proposes Pose-RFT, a reinforcement fine-tuning (RFT) framework for 3D human pose generation from multimodal inputs such as text and images. This work aims to overcome a fundamental limitation in existing pose-specific MLLMs, which is the semantic–spatial alignment gap that arises from their reliance on supervised fine-tuning (SFT) with regression-based objectives.
Instead of directly regressing SMPL parameters via supervised learning, Pose-RFT reformulates the task as a hybrid action space reinforcement learning problem, where the model simultaneously generates discrete language tokens and continuous 3D pose parameters. To achieve stable and efficient optimization in this mixed action space, they introduce HyGRPO, a hybrid reinforcement learning algorithm that applies group-wise reward normalization across sampled responses. Furthermore, the framework incorporates four task-specific reward functions targeting spatial accuracy (for image-to-pose), semantic alignment (for text-to-pose), output format correctness, and embedding-level consistency.
Experiments are conducted on 3DPW, Human3.6M, and RPE for evaluating 3D HPS; on PoseScript for test-topose generation, demonstrating promising performance gains compared with prior SFT-based pose-specific MLLMs, validating the effectiveness of the reinforcement fine-tuning paradigm in achieving both semantic and spatial fidelity in 3D pose generation.

**Strengths:**

1. The paper makes a clear paradigm shift from supervised regression to reward-based reinforcement optimization for multimodal 3D pose generation, which is a timely direction.
2. They adopt HyGRPO to addresses the challenge of jointly optimizing discrete and continuous outputs, which seems to be a reasonable and effective solution. .
3. The integration of task-specific rewards provides a well-structured mechanism to guide policy learning toward both geometric and semantic alignment.
4. Experimental results show solid quantitative improvements over multiple baselines, reinforcing the proposed method’s effectiveness and generality. The ablation studies of HyGRPO show its positive effects on the performance.

**Weaknesses:**

1. Limited qualitative results. Only a few images in Figure 5 and Figure 6 presented in this paper. Providing more visualizations would help readers gain a more intuitive understanding of the performance of the proposed model.
2. Reward sensitivity. Since multiple reward terms are combined, it would be helpful to include a discussion or visualization of how different reward weights affect learning dynamics.
3. Computational overhead. Reinforcement fine-tuning can be computationally expensive; providing efficiency comparisons or scalability analysis would strengthen the practical relevance.

**Questions:**

1. How efficient are the model's inference and training (compared to the supervised training)? This information helps readers understand its advantages and disadvantages.
2. How well the model understands complex textual expressions—is this one of the factors limiting the model's performance?

---

> ### Author Response · Authors · 2025-11-21
> **Rebuttal to Reviewer eur6 - Part 1**
>
> # Summary Response
>
> We sincerely appreciate the reviewer’s constructive feedback and insightful suggestions. Below, we provide point-by-point responses.
>
> ------
>
> > **W1:** Limited qualitative results. Only a few images in Figure 5 and Figure 6 presented in this paper. Providing more visualizations would help readers gain a more intuitive understanding of the performance of the proposed model.
>
> ### **More Qualitative Results:**
>
> We thank the reviewer for this suggestion. We have added more qualitative results for both **text-to-pose (Appendix E) and image-to-pose (Appendix F)** and  in the revised submission to provide a more comprehensive and intuitive understanding of our model's performance.
>
> ------
>
> > **W2:** Reward sensitivity. Since multiple reward terms are combined, it would be helpful to include a discussion or visualization of how different reward weights affect learning dynamics.
>
> ### **Reward Sensitivity and Ablation Study:**
>
> We thank the reviewer for this valuable suggestion. To address the concern regarding reward sensitivity, we first clarify our design philosophy: our framework utilizes **task-specific rewards**, where each reward function is formulated to guide the optimization of a distinct type of output.
>
> To empirically validate this design and assess the importance of each component, we conducted an ablation study where we removed one reward function at a time during RL fine-tuning. Methodologically, for batches corresponding to the removed reward, we **masked the reward signal with a constant scalar (zero)**. This approach preserves the stability of the multi-task training pipeline while strictly isolating the impact of the specific reward function. We have added this experiment to **Appendix G**.
>
> | Model                                | Image-to-Pose (3DPW MPJPE/PA-MPJPE ↓) | Text-to-Pose (PoseScript-H2 $\text{mRecall}^{T2P}$ / $\text{mRecall}^{P2T}$ ↑) |
> | ------------------------------------ | ------------------------------------- | ------------------------------------------------------------ |
> | Baseline (no RL)                     | 91.4 / 59.2                           | 37.4 / 42.0                                                  |
> | w/o Joint Location Reward            | 108.7 / 73.5                          | 55.9 / 60.3                                                  |
> | w/o Semantic Alignment Reward        | 84.0 / 51.3                           | 35.2 / 40.8                                                  |
> | w/o Format Reward                    | 131.9 / 80.6                          | 28.3 / 34.4                                                  |
> | w/o Text Embedding Similarity Reward | 89.8 / 58.7                           | 42.3 / 46.5                                                  |
> | Pose-RFT (Full Model)                | 85.9 / 51.6                           | 53.6 / 57.6                                                  |
>
> The results show that removing each reward leads to a measurable drop in the performance of the corresponding task, confirming that each reward provides essential and non-redundant feedback signals.
>
> ----------
>
> > **W3 & Q1:** Computational overhead. Reinforcement fine-tuning can be computationally expensive; providing efficiency comparisons or scalability analysis would strengthen the practical relevance.
> >
> > How efficient are the model's inference and training (compared to the supervised training)? This information helps readers understand its advantages and disadvantages.
>
> ### **Efficiency Analysis:**
>
> We thank the reviewer for this important question. We provide a detailed comparison of training and inference efficiency between SFT and our Pose-RFT. All experiments were conducted on 4 NVIDIA A100 GPUs.
>
> **Inference Efficiency:** Our RFT approach introduces **zero computational overhead** at inference time. The final model (Pose-RFT) uses the same network architecture as the SFT baseline, just with updated weights. Therefore, the inference speed (e.g., latency and throughput) is identical.
>
> **Training Efficiency:** We compare the computational cost of the SFT stage (5,000 steps) versus our RFT stage (1,000 steps).
>
> - **SFT Stage:** Trained for 5,000 steps with a batch size of 80.
>
> - **RFT Stage:** Trained for only 1,000 steps with a batch size of 16 and a group size of 8.
>
> | Training Stage | Total Training Time (h) | Time Per Sample (s) |
> | -------------- | ----------------------- | ------------------- |
> | **SFT**        | 16.21                   | 0.1459              |
> | **RFT**        | 3.86                    | 0.8688              |
>
> Critically, we wish to clarify that SFT and RFT are complementary stages, not competing alternatives. Following the standard training recipe for modern MLLMs (Pretraining $\to$ SFT $\to$ RFT), our approach adapts this paradigm to the domain of 3D pose generation.

---

> ### Author Response · Authors · 2025-11-21
> **Rebuttal to Reviewer eur6 - Part 2**
>
> ------
>
> > **Q2:** How well the model understands complex textual expressions—is this one of the factors limiting the model's performance?
>
> ### **Impact of Text Complexity**
>
> We thank the reviewer for this insightful question.  To analyze the model's capability in handling complex textual expressions, we used **text length as a proxy for complexity** and evaluated the mRecall performance across different token length intervals on the PoseScript-H2 test set.
>
> | Text Length (Tokens) | mRecall $\uparrow$     | Proportion |
> | -------------------- | ----------- | ---------- |
> | 0-20                 | 34.3%       | 2.84%      |
> | 20-40                | 53.0%       | 21.64%     |
> | 40-60                | 58.0%       | 32.82%     |
> | 60-80                | 56.9%       | 23.74%     |
> | 80-100               | 57.4%       | 13.70%     |
> | 100-120              | 50.8%       | 3.57%      |
> | >120                 | 58.5% (avg) | < 1.7%     |
>
> The analysis suggests that text complexity (in terms of length) is not a primary factor limiting performance.

---

### Author Response · Authors · 2025-11-21
**General Response**

We sincerely thank the reviewers for their constructive feedback and insightful suggestions. We are encouraged that the reviewers recognized the novelty of our paradigm shift from SFT to Reinforcement Fine-Tuning (RFT), the effectiveness of the HyGRPO algorithm in handling hybrid action spaces, and the strong performance of Pose-RFT across multiple benchmarks.

To address the reviewers' common concerns regarding evaluation robustness, reward component contribution, and the effectiveness of RFT for visual HMR, we have conducted extensive new experiments. Below is a summary of the key updates:

**1. Robustness of Evaluation** To address concerns regarding the entanglement of reward and evaluation models (Reviewer kH8o, Reviewer 6Rpp), we introduced a Cross-Retriever Evaluation using an independent retrieval model (*PoseEmbroider*) in Appendix H. The results confirm that our improvements stem from robust semantic alignment rather than overfitting to a specific feature space.

**2. Reward Component Ablation** To clarify the impact of individual reward terms (Reviewer eur6, Reviewer kH8o, Reviewer 6Rpp), we added a comprehensive Reward Ablation Study in Appendix G. The results demonstrate that each task-specific reward (Spatial, Semantic, Format, Similarity) is essential and non-redundant.

**3. Application to Generative Visual HMR Methods** To demonstrate the generality of our approach (Reviewer rGEb), we applied our RFT framework to a generative visual backbone (*ScoreHypo*). The results show that RFT significantly improves even strong generative baselines, confirming that our method is complementary to, not redundant with, visual HMR methods.

**4. Additional Analysis & Visualizations**

- **Efficiency:** We added a detailed training/inference efficiency comparison between SFT and RFT (Response to Reviewer eur6 and Reviewer 6Rpp).
- **Qualitative Results:** We expanded visual results for both text-to-pose and image-to-pose tasks, including video stability analysis, in Appendix E & F (Response to Reviewer eur6, Reviewer rGEb).

We hope these revisions and our detailed point-by-point responses satisfactorily address your concerns.

---

### Author Response · Authors · 2025-11-26
**Follow-up on Rebuttal Response**

Dear Reviewers,

Following up on our response posted a few days ago, we wanted to kindly check if the provided clarifications and new results have sufficiently addressed your comments.

We value your feedback and are eager to engage in further discussion to clear up any remaining concerns before the decision phase. Please let us know if there is anything else we can clarify.

Best regards, The Authors

---

### Author Response · Authors · 2025-12-03
**Summary Response to Area Chair and Reviewers**

We sincerely thank the Area Chair and all reviewers for their time and effort throughout this process. While we regret that the recent information leak incident prevented us from receiving the reviewers' feedback on our rebuttal, we believe that our responses and additional experiments have effectively addressed the concerns raised.

**To Reviewer eur6:**

- **W1:** We have added extensive **qualitative results** in Appendix E and Appendix F to visually demonstrate our model's performance.
- **W2:** We conducted a detailed **Reward Sensitivity and Ablation Study** in Appendix G, which quantifies the impact of each reward term on RL training and confirms that our current configuration is optimal.
- **W3 & Q1:** We provided a comprehensive **comparison of training and inference efficiency** between RFT and SFT to offer a clearer understanding of our method's computational cost.
- **Q2:** We performed an experiment using text length as a proxy for complexity, demonstrating that **text complexity** is currently not a primary factor limiting model performance.

**To Reviewer rGEb:**

- **W1:** To address the **major concern regarding the necessity of RFT for visual HMR**, we applied RFT to a generative visual-only HMR backbone (*ScoreHypo*) and achieved significant performance gains, nearing the performance of SOTA specialist models. This confirms that our RFT paradigm is effective and complementary to visual HMR methods.
- **W2:** We provided a detailed explanation of the architectural challenges involved in integrating existing **discrete pose tokenizers** into causal MLLMs, thereby clarifying the motivation for adopting our HyGRPO approach.
- **W3:** We commit to including full evaluation results on the **EMDB benchmark** in the camera-ready version.
- **Q1:** We included additional experimental **results on videos** in Appendix F.

**To Reviewer kH8o:**

- **W1 & Q1:** We reaffirmed our contribution as a Unified Pose Generation framework that offers unique **Reasoning Pose Estimation** capabilities compared to traditional methods. Regarding the performance gap, we conducted a decomposition experiment to analyze the trade-off between generalist capabilities and specialist performance across training stages.

  We demonstrated that our "Visual Specialist" (Stage I) achieves competitive accuracy (35.6 mm PA-MPJPE). Consequently, we proposed an **Integrated Pose System**: since the Specialist and Generalist heads share the same visual backbone, our framework can adaptively switch between them—leveraging the Specialist for maximal visual precision and the Generalist for interactive reasoning—thereby bridging the performance gap without sacrificing unique MLLM capabilities.

- **W2 & Q2:** We added a **Cross-Retriever Evaluation** using an independent model (*PoseEmbroider*) in Appendix H, proving the robustness and generalization of our model.

- **W3:** We clarified the distinctions between our approach and **previous mainstream Hybrid RL methods** to further highlight our algorithmic novelty.

- **W4:** We completed a **Reward Sensitivity and Ablation Study** (Appendix G) to validate the necessity and optimality of our reward settings.

**To Reviewer 6Rpp:**

- **W1 & Q1:** We conducted a detailed **Reward Ablation Study** (Appendix G) to demonstrate the specific impact of each reward term, confirming that our current settings are optimal.
- **W2 & Q2:** We included the **Cross-Retriever Evaluation** in Appendix H to demonstrate the generalization of our method.
- **W3:** We provided the **efficiency comparison** (training / inference) between RFT and SFT.
- **Q3:** We conducted new experiments analyzing the **impact of Group Size $G$** on performance, efficiency, and stability, confirming that $G=8$ is the most reasonable setting.

---

### Meta-Review · Area_Chair_nQXx · 2026-01-07

**Summary:**

This paper presents a reinforcement fine-tuning (RFT) framework for 3D human pose generation from multimodal inputs such as text and images. The key technical element is HyGRPO, designed for stable optimization under this mixed discrete-continuous output setting, together with task-specific reward functions. This is the first work along this direction. Three reviewers have recognized its technical contribution; one reviewer questioned the motivation for the hybrid design and the necessity of proposing HyGRPO. The authors have provided a reasonable explanation for this particular design, and it is believed that this is not a trivial problem. Therefore, the final recommendation is Accept.

**Reviewer Concerns:**

Addressed: 1) more qualitative results: The authors added expanded qualitative results and video-based analysis in the appendix; 2) Ablation of reward components: A reward ablation study was added and used to argue that each reward term is non-redundant; 3) reward hacking: The authors added a cross-retriever evaluation using an independent retrieval model; 4) Is RFT necessary? : The authors added an experiment applying RFT to a generative visual HMR backbone and reported consistent improvements.

Not fully addressed: 1) Evaluation on EMDB: results are not provided in the rebuttal. 2) Performance gap vs. specialist HMR models on standard metrics remains.

**Reviewer Scores:**

Reviewer eur6 might keep the initial rating, given the score is already 8; Reviewer rGEb might increase the score to 6, given that the main concern about the motivation for hybrid design is well articulated; Reviewer kH8o might stay with the initial rating given that the concern about performance gap cannot easily be addressed;  Reviewer 6Rpp might increase the score to 8 since all the concerns seem to be addressed properly.

---

### Decision · Program_Chairs · 2026-01-26

Accept (Poster)